# Deep learning predicts real-world electric vehicle direct current charging profiles and durations

Siyi Li[1] ✉, Mingrui Zhang [1], Robert Doel [2], Benjamin Ross [2] & Matthew D. Piggott[1]

Accurate prediction of electric vehicle charging profiles and durations is critical for adoption and optimising infrastructure. Direct current fast charging presents complex behaviours shaped by many factors. This work introduces a deep learning framework trained on 909,135 real-world sessions, capable of predicting charging profiles and durations from minimal input with uncertainty estimates. The model initiates predictions from a single point on the power and state-of-charge profile and incrementally refines them as new observations arrive, enabling real-time updates. The model generalises across vehicle types and charging scenarios. It achieves 90% accuracy in predicting charging duration from a single point, and 95% accuracy with an absolute error under one minute using six points within five minutes. This work shows that using readily available input data at charge time enables accurate prediction of charging behaviour and offers a practical, scalable solution for deployment, energy planning, and infrastructure reliability.

The electrification and decarbonisation of transportation are critical components of the global strategy towards combating climate change, reducing greenhouse gas emissions and promoting more sustainable energy solutions. Central to this effort are electric vehicles (EVs) with lithium-ion batteries (LiBs), which have become increasingly viable due to rapid technological advancements and increasing cost efficiency[1–3]. In 2023, electric car sales comprised approximately 20% of total car sales worldwide, and projections for 2024 indicate that EV market share will increase to over 11% in the United States, 25% in Europe, and 45% in China[4]. Despite these promising trends, the widespread adoption of vehicle electrification depends on the simultaneous development of accessible, reliable and affordable public charging infrastructure. The IEA's Announced Pledges Scenario (APS) projects that the global number of public charging points will exceed 25 million by 2035, a sixfold increase relative to 2023, which is expected to facilitate the transition to electric vehicles for mass-market consumers. This expansion is crucial because the lack of adequate charging infrastructure remains a significant barrier to EV market growth and a major source of consumer anxiety, affecting the perceived flexibility and convenience of EV use[4–8].

Direct current fast charging (DCFC) stations play an important role in addressing range anxiety and facilitating long-distance travel by significantly reducing charging times. Capable of charging an EV battery from 20 to 80% in just 20 to 30 minutes under ideal conditions, DCFC stations are indispensable for drivers needing quick recharges during long trips or those without home charging access[9]. The deployment of DCFC infrastructure is essential for increasing EV integration, particularly in urban environments and along major transportation corridors where high traffic volumes demand rapid charging solutions. However, the complex dynamic nature of the charging profiles and the resulting fluctuations in power demand present challenges in station management and grid integration, complicating their incorporation into the electric mobility ecosystem.

The accurate prediction of charging profiles can enhance user experience by providing accurate estimates of charging durations, as well as providing real-time information on charger availability and expected wait times. For EV fleet managers in various sectors, including public transportation, emergency services, logistics, or mining, this capability facilitates real-time decision-making for vehicle

[1]Department of Earth Science and Engineering, Imperial College London, London, UK. [2]Shell Research Limited, London, UK. ✉ e-mail: siyi.li20@imperial.ac.uk

deployment, thereby improving operational efficiency and reducing idle time[10–12]. Moreover, charging profile prediction could help grid operators better estimate the total energy demand for each charging session, enabling more accurate forecasting of electricity demand and more efficient management of grid resources. This capability can also potentially benefit Vehicle-to-Grid (V2G) technologies by better determining when and how much energy can be drawn back from EVs to support the grid during peak demand, thereby enhancing grid stability and optimising energy use[13,14].

Recent advances in data-driven modelling of lithium-ion battery systems have increasingly addressed challenges associated with random, scarce, and heterogeneous data using sophisticated machine learning techniques[15–19]. These studies, often based on real-world datasets, have achieved strong performance in tasks such as state-of-health estimation, lifetime prediction, and retired battery sorting, and represent meaningful progress toward the development of robust and generalisable battery analytics. By contrast, research specifically focused on predicting EV charging profiles remains comparatively limited. Existing work in this area has largely relied on relatively small datasets, often collected under laboratory or semi-controlled conditions using instrumented cells or proprietary battery management system (BMS) data[20–24]. Such data are frequently tied to specific vehicles and may not be publicly accessible due to design constraints and privacy concerns specific to original equipment manufacturers (OEMs), which limits the broader applicability and scalability of these approaches. Moreover, while the use of real-world EV data is becoming more common, it has predominantly been applied to tasks such as state-of-health estimation, fault detection, or lifetime prediction[25–29], rather than to the forward-looking task of charging profile prediction, despite its relevance for smart charging, range forecasting, charger utilisation planning, and energy services.

The success of deep learning in various domains, such as computer vision[30], natural language processing[31], drug discovery[32,33], wind energy[34–36] or climate and weather forecasting[37,38], underscores the potential of these techniques to address complex predictive tasks when supported by large-scale, high-quality datasets. In this context, this work introduces a deep learning framework that is built on 909,135 real-world DCFC charging sessions, encompassing a wide variety of EV models and charging scenarios.

In this work, a deep learning workflow is presented for predicting EV DC fast-charging profiles in real-world settings, demonstrating reliable performance while prioritising practicality and ease of deployment through the exclusive use of data that is readily available and accessible at charge time. The proposed work flow comprises an anomaly detection model that identifies irregular charging behaviours, potentially stemming from malfunctioning charging infrastructure or vehicle-related anomalies, and a charging profile prediction model that can generate predictions from as little as a single data point on the power/state-of-charge charging profile, while also offering uncertainty estimates and dynamically updating predictions in real time as charging progresses and more data becomes available. The developed models were tested in a real-world operational scenario with previously unseen charging data collected from a period later than the training data; results showed that the model can accurately predict real-world EV DCFC charging profiles and their associated charging times, even with minimal inputs. Specifically, the model achieves 90% relative accuracy in predicting charging time with a single point, and 95% relative accuracy and absolute error of less than one minute with six points collected in five minutes or less. This work demonstrates the efficacy of deep learning models in predicting EV charging profiles using minimal input data while emphasising the practicality of the proposed workflow.

## Results
### DC fast charging complexities and challenges with designing predictive models
Accurately predicting the charging profile, defined as the charging power as a function of the EV battery state-of-charge (SoC), of real-world EV DCFC sessions is inherently challenging due to the complexity and variability of the charging process and the multitude of influential factors involved. The same EV tends to exhibit different charging behaviours when the charging process is initiated at different SoCs or with connectors of different power ratings, with the batteries generally charging faster at lower SoCs and slowing down significantly when approaching full capacity. The temperature of the battery also plays an important role in the overall charging speed, with lower temperatures often leading to reduced charging rates. Moreover, the different EV models each with particular constraints imposed by the vehicle's hardware and battery management systems (BMS), often exhibit remarkably distinct charging profiles. Many other factors, including the battery state-of-health (SoH), different connector types, and grid conditions, also affect the charging profile.

To effectively analyse and comprehend these diverse charging behaviours, it is crucial to employ an extensive dataset that encompasses the variability in charging patterns across various conditions and EV models. In this work, a carefully curated dataset of 909,135 high-quality DCFC charging sessions with minimal missing data was chosen for analysis, from charging data collected from 612 different DCFC chargers located in northwestern Europe between November 2021 and July 2024. Each session includes time series data for state-of-charge and charging power, as well as detailed information including connector power rating, connector type, session duration, total energy delivered, and the geographical location of the charging station.

An overview of the DCFC charge curve dataset used in this work is presented in Fig. 1. This comprehensive dataset includes a diverse range of real-world EV charging profiles collected from numerous makes and models, charged with connectors rated from 50 kW up to 360 kW. Figure 1d shows that while stations with higher power ratings are predominantly utilised by EVs with larger battery capacities, there are also many cases where large-battery EVs use connectors rated below 75 kW, and EVs with capacities under 50 kWh use connectors rated above 300 kW. As evident in Fig. 1e, there is substantial variability in charging sessions both in terms of the percentage of SoC added and the corresponding session duration, which limits direct comparison with laboratory references. The majority of sessions begin between 10% and 60% SoC and end near full capacity (at least 80%), but the dataset also includes a significant number of shorter top-up events. These different charging habits likely reflect diverse user preferences and requirements. Some drivers charge close to full capacity out of range anxiety, while others interrupt charging earlier due to time constraints or perceptions of battery health. Charging duration is equally varied, ranging from just over 10 minutes for brief top-ups to more than an hour when charging close to full capacity. It can be clearly observed that charging slows markedly as the battery approaches full, with sessions targeting more than 90% SoC taking much longer than those ending between 80 and 89%, even from the same starting point. Sessions resulting in less than a 10% change in SoC were excluded from the analysis.

Complementary insight is provided by Fig. 2, which focuses on representative charging profiles. Figure 2a shows reference DC charging curves for several EV models measured under ideal laboratory conditions at Shell Technology Center. In these profiles, supplied power typically increases or remains constant until a threshold SoC is reached, after which it progressively declines to zero. Different vehicle–battery systems exhibit characteristic behaviours at different rated connector powers. Figure 2b illustrates how real-world profiles differ: each subplot represents charging by a single anonymous user with the same connector type and power rating but under different

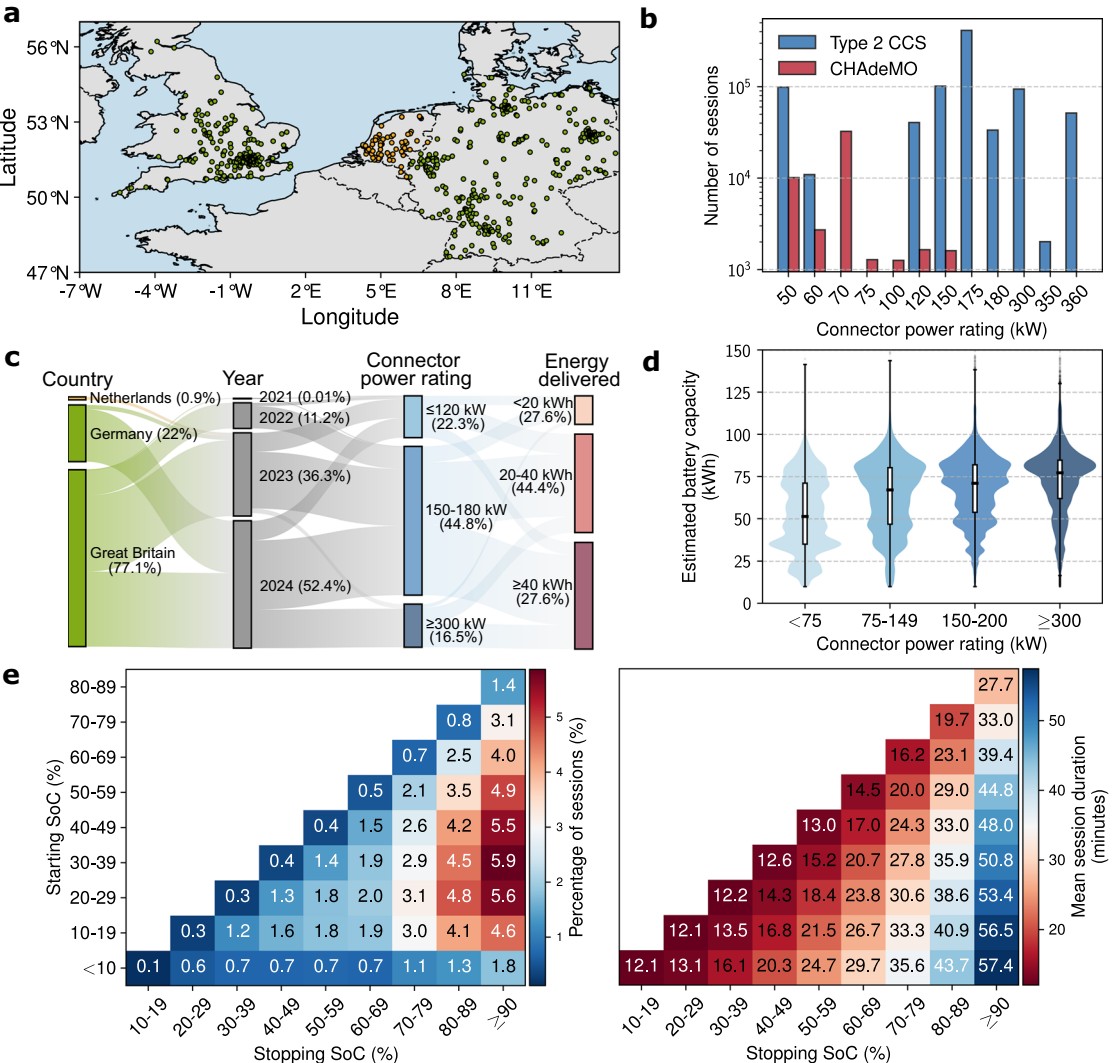

Fig. 1 | The dataset utilised in this work encompasses a wide variety of electric vehicles (EVs) with different battery capacities, chemistries, and battery management systems (BMS), capturing charging behaviour under real-world and often sub-optimal conditions. a The geographic distribution of the charging stations from which the charge curve dataset was collected. Stations located in the Netherlands did not report cumulative energy delivered and are therefore excluded from the quantitative analyses in panels (b–e). b A diverse array of charging stations with rated connector power ranging from 50 kW to 360 kW are included in the dataset, with the primary connector types being Type 2 Combined Charging System (CCS) and CHAdeMO. c Most charging sessions analysed in this study originated from Great Britain and Germany, with fewer than 1% of sessions coming from the Netherlands. The dataset was collected from 2011 to 2024, and covers a large range of sessions with various levels of delivered energies and connector power ratings. It should be noted that these describe only the subset of data examined here and should not be taken as representative of Shell's overall charging operations. d The distribution of estimated battery capacity across different connector power ratings. Violin plots show the data distribution (kernel density), with embedded box plots indicating the median (centre line), interquartile range (box limits), and whiskers (1.5 × interquartile range). e The percentage of sessions as a function of starting and stopping state-of-charge (SoC), as well as the average session duration of these sessions. Note that for plotting purposes, these are computed from a fraction of the dataset due to minor inconsistencies in recorded starting or stopping SoC values in some sessions.

ambient temperatures. Even for one EV, charging patterns are highly variable, depending on factors such as starting SoC and external conditions. While battery temperature itself is not recorded at stations, ambient temperature at the charging location and time was obtained from ERA5 reanalysis data[39] and used as a proxy. The influence of temperature is evident, with colder conditions producing slower charging. The extended collection period and wide geographical distribution of stations captured sessions across a broad climatic range, from −14 °C to 35 °C.

The analysis of the DCFC dataset indicates that an ideal predictive model for EV charging profiles should possess several key capabilities. It should be able to generalise across different EV models as well as support different connector types and power ratings. The model should be able to capture the effects of influential variables such as starting SoC, battery capacity and environmental conditions on charging behaviour, while relying only on inputs that can be readily obtained by the charging station in a practical manner. Additionally, it should also be able to efficiently process large, noisy real-world datasets and handle charging profiles of varying lengths. In order for it to have more practical application, the model should also be able to make reliable predictions with minimal input data and to make real-time updates to its predictions as new data is received. Furthermore, it must be able to effectively manage uncertainties related to various factors, including types of EV and BMS, battery SoH, and grid conditions, which are factors that are not directly observable or accessible at charge time. This work presents a deep learning framework specifically designed to meet these criteria, offering a practical solution for predicting charging profiles that

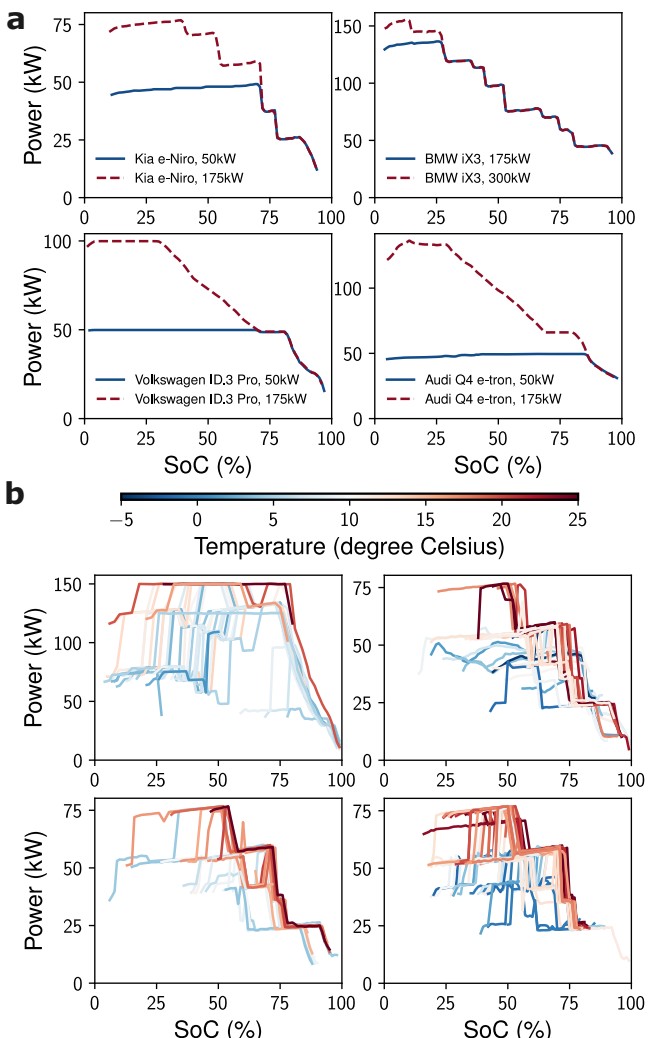

**Fig. 2 | Comparison between controlled laboratory and real-world electric vehicle direct current fast charging profiles. a** Reference charging profiles of Kia e-Niro, BMW iX3, Volkswagen ID.3 Pro, and Audi Q4 e-tron measured in controlled laboratory sessions under ideal charging conditions of ambient temperature of 30 °C with Combined Charging System (CCS) chargers of different rated connector powers. **b** Each subplot represents the charging profiles of a single anonymous user (and likely the same EV) using the same connector type and power rating under different environmental ambient temperatures. The sampled charging profiles demonstrate that real-world charging patterns are highly diverse and irregular, and dependent on many factors including starting SoC and ambient temperature.

dynamically adapts to real-time data while addressing the uncertainties inherent in EV charging scenarios.

## Anomaly detection and time series forecasting with β-variational autoencoder and temporal fusion transformer

The deep learning workflow is designed for seamless integration with real-world deployment, with the model inputs consisting of anonymised information from charging stations. It contains two main components, a β-VAE-based anomaly detection model and a charging profile prediction model. Input charging profile data are structured as multivariate time series of charging power and SoC, augmented with static covariates including starting SoC, connector power rating, connector type, estimated EV battery capacity, and ambient temperature. While the anomaly detection model requires complete charging profiles to establish normative behaviour, the prediction model operates on partial profiles during real-time deployment, enabling adaptive forecasting. The static covariates selected are

readily accessible: the charger provides information on starting SoC as well as connector type and power rating. Battery capacities of the EVs can be estimated dynamically by the charger as charging progresses or could be obtained through image recognition used to identify the EV model, while ambient temperature could be directly measured. However, in this work, battery capacities were estimated using the total energy delivered during each session and the total change in SoCs, while ambient temperatures were obtained from ERA5 reanalysis data as described above. It is important to note that although the workflow is optimised for the inclusion of estimated EV battery capacity and ambient temperature, it remains capable of delivering reliable results without these features, with only a slight reduction in accuracy. Further details are provided in Supplementary Fig. 6. An illustration of the proposed workflow, the anomaly detection model, and the charging profile prediction model are shown in Fig. 3.

Large-scale real-world EV charging data are inherently noisy and inevitably contains spurious or corrupted measurement data, potentially rooting from a large number of sources, including connector misfits, firmware interruptions, charger malfunctions, data recording errors, or issues within the EV itself. While downstream predictive models must be robust to typical noise in real-world data, clearly abnormal sessions could lead to biases during both training and inference if not properly addressed. As these faults are not explicitly labelled in practice, a beta-variational autoencoder (β-VAE)-based anomaly detection model is trained on the complete set of full charging profiles in order to filter out spurious sessions and limit the influence of outliers. The β-VAE model is detailed in Fig. 3b, and was trained to learn a compressed latent representation of normal charging profiles by minimising both the reconstruction loss and the Kullback-Leibler (KL) divergence loss. After training, each session is assigned a reconstruction-error score quantifying its conformity to the learned manifold, abnormal charging profiles are then detected as instances where the reconstruction loss exceeds a statistically derived threshold, signalling deviations from standard charging behaviour. Instances of detected abnormal charging sessions are presented in Fig. 4a.

Following outlier removal via the anomaly detection model, the charging profile prediction model was trained on the filtered dataset with a self-supervised learning approach, which involves predicting full charging profiles from partially available data, with varying segments of the charging profile intentionally concealed during training. This training strategy allows the model to handle inputs with arbitrary amounts of missing data, providing the model with the capability of making predictions with inputs as minimal as a single point on the charging profile, and adjusting its predictions in real-time as new charging data becomes available, as shown in Fig. 4b. Additionally, the use of the quantile loss function enables the model to generate probabilistic predictions that are crucial for anticipating a wide range of potential charging behaviours with limited input, enhancing the model's robustness and deepening the understanding of diverse charging scenarios. Figure 4b also demonstrates the model's ability to manage prediction uncertainty in complex charging profiles, where the predictions' quantile ranges effectively capture the variability in potential outcomes. It should be noted that the examples shown represent more challenging charging profiles with relatively high initial uncertainty that decreases as more data becomes available. Model performance tends to be significantly better in simpler, more common cases, where the quantile ranges are often much narrower, even with minimal input data.

Regarding the model architectures, both the anomaly detection model and the charging profile prediction model incorporate components from the temporal fusion transformer (TFT), which is a state-of-the-art time-series forecasting model originally developed by Lim et al.[40]. In both models, the static covariates are not only utilised as inputs but also encoded as context vectors to influence temporal

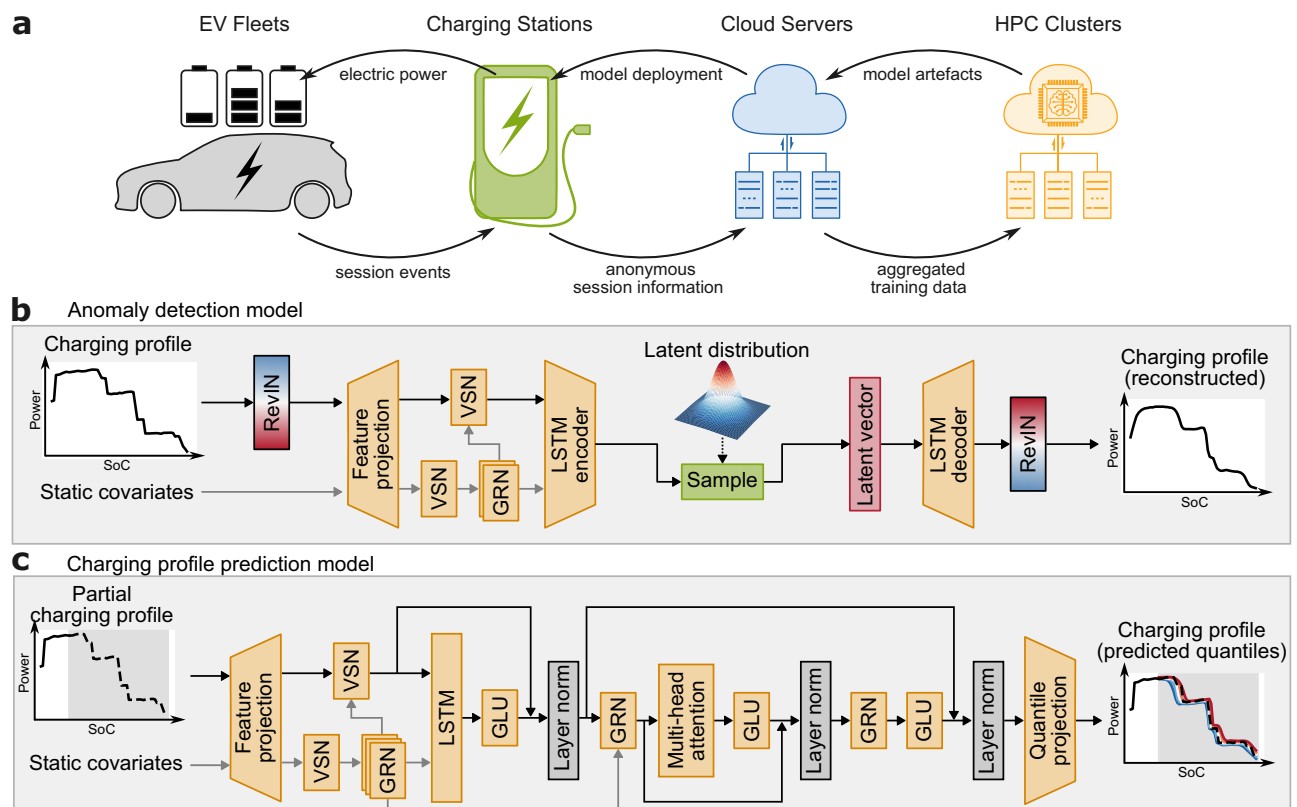

**Fig. 3 | Overview of the developed deep learning framework. a** A high-level illustration of the AI-based system for direct current fast charging (DCFC). Charging session data are continuously collected from charging stations, aggregated on cloud servers, and delivered to high-performance computing (HPC) platforms for large-scale model training. The resulting trained models are deployed via cloud servers to provide real-time charging profile and time predictions, together with anomaly detection. **b** A schematic of the beta-variational autoencoder ($\beta$-VAE)-based anomaly detection model, which identifies irregularities in charging sessions by reconstructing the original power/state-of-charge (SoC) charging profile and evaluating the associated reconstruction error. RevIN denotes reversible instance normalisation, and long short-term memory (LSTM) is used for sequence modelling. **c** A schematic of the probabilistic charging profile prediction model, which predicts the quantiles of a full charging profile from partial inputs. Gated linear unit (GLU), gated residual network (GRN) and variable selection network (VSN), are key architectural components of the anomaly detection and charging profile prediction models, with additional details provided in Supplementary Fig. 1.

dynamics. The integration of variable selection networks enables adaptive weighting of these covariates, effectively learning to prioritise the most significant variables and also enhancing model interpretability. In the anomaly detection model, reversible instance normalisation (RevIN) layers address distributional shifts in charging profiles, ensuring consistent feature scaling across diverse sessions[41]. Additionally, sequence-to-sequence long short-term memory (LSTM) layers in both models capture sequential dependencies in the time series data. Multi-head attention layers in the charging profile prediction model allow it to focus adaptively on different parts of the input sequences, further improving both interpretability and predictive power[31].

**Time to reliable prediction operationally post-plug-in**
Abnormal sessions were identified through reconstruction errors in the top percentile of the distribution and through cases where reconstructed charging durations exceeded actual values by more than 15 minutes. The temporal criterion specifically addresses pathological cases where power measurements transiently drop to zero, typically indicating sensor faults or disconnection events, which could artificially inflate predicted charging times and unfairly penalise model performance. The exclusion of these anomalous sessions ensures that training and evaluation reflect only physically plausible charging behaviour while preserving the overwhelming majority of statistically meaningful data. This minimal exclusion of 9334 (1.02% of the full dataset) was carefully selected to remove only clear outliers without

affecting the underlying data distribution. Following this anomaly exclusion step, the charging profile prediction model was trained on the filtered dataset and subsequently evaluated on two independent, previously unseen test sets. The first test set consists of contemporaneous sessions collected within the same time frame as the training data, whereas the second test set comprises sessions collected immediately following the period over which the training data was collected, and was designed to simulate real-world operational conditions and assess the model's predictive performance in a near-future context. Distributional differences between the training and test sets are quantified and visualised in Supplementary Figs. 3 and 4 using latent representations from the $\beta$-VAE model.

The performance of the charging profile prediction model is detailed in Fig. 5. All results reported correspond to an ensemble model constructed by taking a weighted mean of predictions from three independently trained instances of the TFT model, each initialised with different random seeds and data splits. Although the best-performing individual model achieved slightly higher accuracy on certain metrics, the ensemble was selected for analysis as it provides greater robustness and is more representative of typical model behaviour under varied conditions. Training and validation loss curves for the individual training runs of the TFT model are detailed in Supplementary Fig. 2.

The model's proficiency in both predicting the charging profiles and estimating the corresponding charging times is evaluated for the two test sets and reported in Fig. 5a, where the median predicted

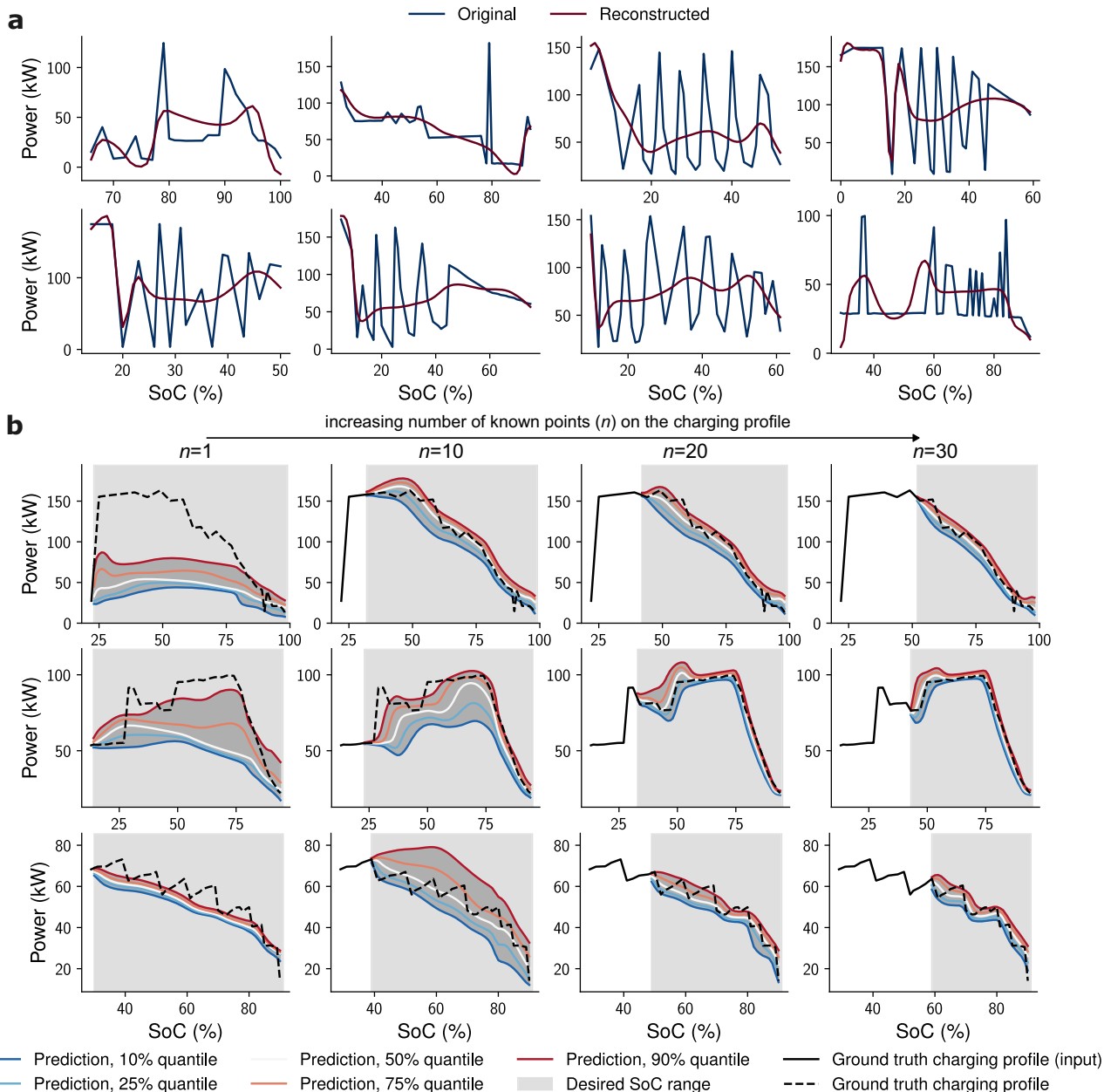

**Fig. 4 | Results from the anomaly detection and charging profile prediction models. a** Examples of irregular power/state-of-charge (SoC) charging profiles detected by the anomaly detection model, with reconstruction errors within the top 1% or where the computed charging time from the reconstructed charging profile differed from the ground truth by more than 15 min. **b** Quantile predictions for three sessions (top, middle, bottom rows) from the charging profile prediction model using varying numbers of known points on the charging profile as input.

charging profiles and their associated charging times are compared against ground truth values. It can be seen that while the model performed marginally better in test set one than in test set two, the difference is relatively small. Overall, the model achieved an average relative accuracy of approximately 90% for both test sets when charging time is predicted from a single input point. The corresponding mean absolute error of less than 2.5 minutes is reported separately in Supplementary Fig. 5. This value is the lower bound of the model's performance in predicting charging time using relative error, with performance also assessed using absolute-error metrics such as MAE in charging time and charging curve accuracy across input lengths from 1 to 15 points. Accuracy increases progressively as more data becomes available, and even this lower-bound accuracy can enable operationally useful actions such as anticipating charger availability, assessing session completion within certain time windows, and

providing early completion time estimates to drivers. Even with this minimal input, the model demonstrated significant predictive power and explained 91% of the variance in charging time predictions, as can be seen in Fig. 5b. As the number of known points on the charging profile increases, the model is able to adapt quickly and update its predictions, improving its accuracy in predicting charging profiles and charging times while narrowing its estimation of uncertainties. As shown in Fig. 5c, d, the proportion of sessions with high prediction uncertainties decreases sharply from 15% with a single input point to less than 2% when five or more points on the charging profile are available. The prediction for a session is deemed uncertain if the difference in charging time between the 10% and 90% quantile predictions exceeds 20 minutes or 50% of the median predicted charging time. With six known points on the charging profiles, which correspond to five minutes of charging or less, 90% of sessions in both test

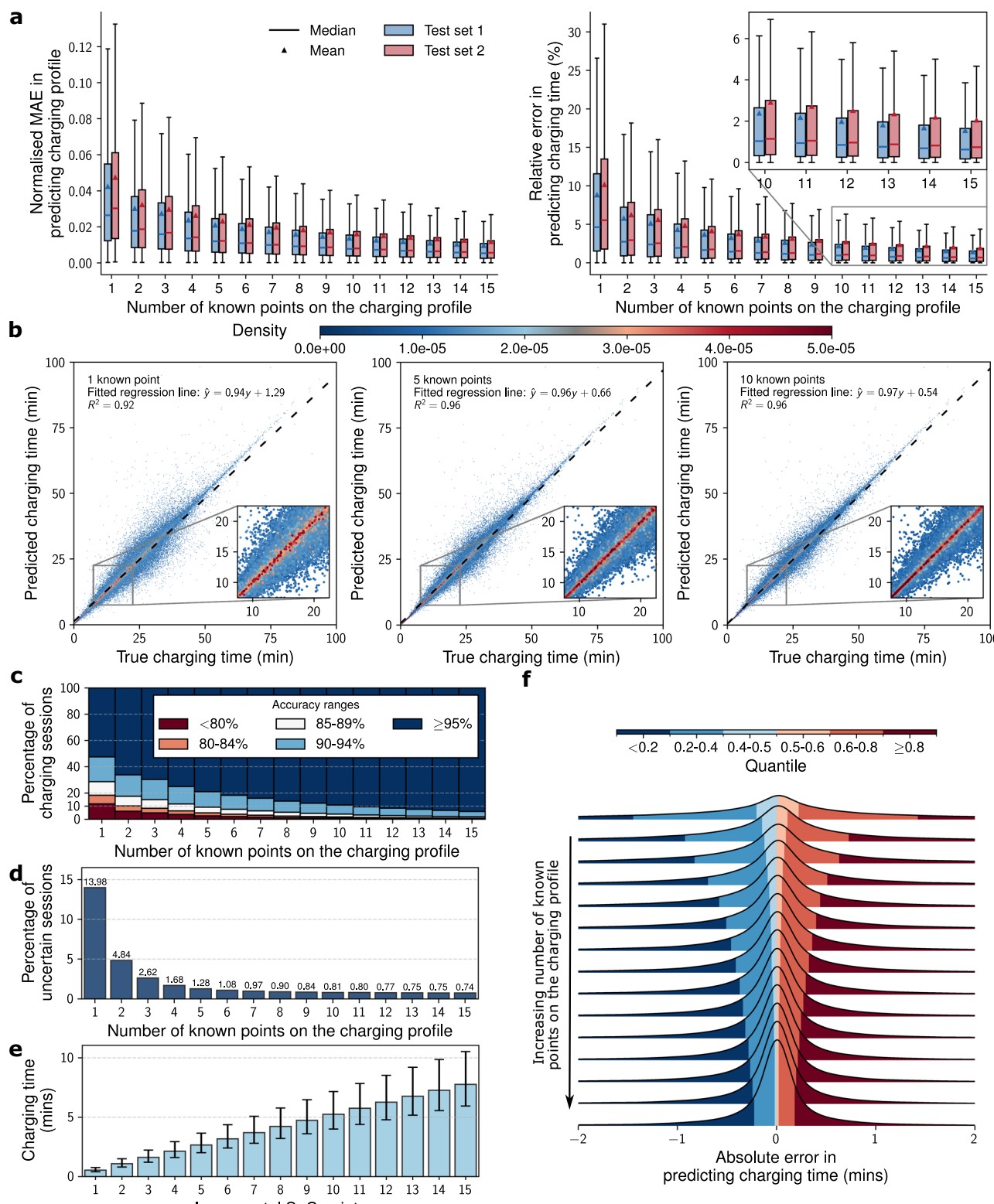

sets achieve more than 90% accuracy in predicting charging time. When 15 points are known on the charging profile, only 1.45% of sessions fall below 90% accuracy in charging time prediction, and just 0.79% fall below 80%.

It should be noted that while the charging profile data are logged by chargers at regular intervals of one minute, depending on the provider, the machine learning model is designed to update its predictions in an event-driven manner, specifically when the reported SoC

increases by at least 1%, reflecting the integer precision defined by the Open Charge Point Protocol (OCPP) standard. Many operators commonly use a 60-second logging interval to balance data volume and utility, but this setting is configurable. For instance, busier or critical sites could report more frequently to enable more accurate predictions. Since DC fast charging is highly non-linear, the timing of these SoC updates is irregular and varies across sessions. Figure 5e shows the median elapsed time required to accumulate a given number of SoC

**Fig. 5 | Performance evaluation of the charging profile prediction model, where the model's median predictions were compared to ground truth charging profiles and times.** Charging times were calculated based on either the completion of the charging session or reaching 80% SoC, whichever came first. Analysis was confined to sessions with a strictly greater than 15% change in SoC. **a** The charging profile prediction model's performance on predicting the power-SoC charging profile and charging time for different numbers of known points of the charging profile as input, evaluated on the two test sets. Normalised mean absolute error (MAE) was computed as the MAE of predicted power normalised by connector power rating. Box plots show the median (centre line), mean (triangles), interquartile range (box limits), and whiskers (1.5 × the interquartile range). **b** Scatter plot with density overlays comparing median predicted charging times against ground truth across different numbers of input points. The results combine data from both test sets. **c** The accuracy ranges of the test set sessions, evaluated on both test sets. **d** The percentage number of uncertain sessions for different numbers of input points across both test sets. **e** The median time required to accumulate a specific number of points on the charging profile, with error bars representing the interquartile range. **f** Distribution analysis of absolute errors in predicting charging time, across varying numbers of input points on the charging profile, incorporating results from both test sets.

points. Early in the charging process, multiple SoC updates are typically observed within a few minutes, while in later stages, longer durations are required due to reduced charging power. This results in a variable effective temporal resolution, governed by the charging dynamics rather than fixed time intervals. In practice, model updates may occur every minute in the early stages of a session, enabling the model to make timely and increasingly accurate predictions as more data becomes available. As the charging rate slows and updates become less frequent, the model's predictions have typically already converged, reducing the urgency for further refinement. Figure 5f shows the distribution of absolute errors, while relative errors are presented in Supplementary Fig. 5c. It can be observed that when the number of input points is low, the model is marginally biased towards underestimating charging time. As more data is received, the model consistently refines and improves prediction accuracy. The distributions of both relative and absolute errors become increasingly narrow, with the vast majority of sessions ultimately achieving a prediction error of within ± 5% and an absolute error of less than one minute. It is worth noting that sessions that consistently exhibit high uncertainty, regardless of the number of input points, may indicate abnormal charging behaviour that was not filtered out by the anomaly detection model. This persistent uncertainty detected during operational inference time can serve as a real-time feature for flagging anomalies to customers, thus providing an additional layer of anomaly detection during the model's deployment.

## Model-derived insights into real-world charging behaviours

A more transparent interpretation of the inner workings of machine learning models could provide valuable insights into EV DC charging behaviours. This is accomplished by interpreting the anomaly detection model through visualisation of the learned low-dimensional latent embeddings and examining the charging profile prediction model by analysing the influence of static covariates on its predictions. To focus on more representative charging sessions, the real-world charging data were first screened and compared against a set of reference charging profiles. The reference curves, generated through experiments conducted by Shell under controlled laboratory conditions, encompass a diverse range of EV brands, models, connector power ratings, connector types, and ambient temperatures. A curve-matching software implemented by Shell was used to identify the closest matches between the real-world profiles and the reference curves. Around 15 thousand charging profiles with more than 60 data points and exhibiting a strong match with a reference curve were subsequently compressed into latent representations by the $\beta$-VAE encoder, projected onto a 2D place using t-distributed stochastic neighbour embedding (t-SNE), and visualised in Fig. 6a. Distinct clusters emerge within the latent space, with EVs of the same brand and model generally grouped closely together. By clustering real-world charging sessions with similar reference profiles, the $\beta$-VAE encoder effectively preserves the relationships defined by the curve-matching process and captures meaningful patterns in the data. The grouping for Nissan Ariya/Leaf is less obvious, due to overlapping reference profiles with other brands, as detailed in zoomed-in views of selected clusters shown in Fig. 6b. With the introduction of RevIN layers in the

$\beta$-VAE model, it successfully groups together charging profiles that share the same shape but differ in scale, as demonstrated by the unified clustering of Volkswagen ID.3 Pro and Pro S charging sessions. This scale-invariant behaviour arises from RevIN's ability to normalise input dynamically across sessions, allowing the latent space to prioritise shape-related features over absolute magnitude, which is essential when dealing with heterogeneous charging infrastructure and battery capacities.

Figure 6c presents both a baseline comparison and an ablation study of model architecture. The TFT achieves the lowest normalised MAE among all models, outperforming recurrent neural networks (RNN), gated recurrent units (GRU), LSTM and the vanilla Transformer when evaluated on a combined dataset of test sets one and two. To isolate the contributions of key architectural components, two reduced variants are evaluated: removing the multi-head attention and feed-forward linear modules yields the variable selection network plus LSTM (VSN-LSTM), while removing the recurrent components reduces the model to a pure Transformer. The superior performance of TFT over both simplified variants indicates that attention and recurrence provide complementary advantages for learning meaningful representations of charging profiles. Given the self-supervised nature of the task and the variability in input sequence lengths, comparison with traditional statistical learning approaches is not applicable, as such methods typically assume fixed-length feature vectors and supervised targets. TFT was evaluated across three independent random seeds and data splits, despite the computational cost of 72 NVIDIA A100 GPU hours per training run. This robustness check was applied selectively to this model to validate the stability and generalisability of its performance, whereas the baselines were evaluated once under fixed conditions for efficiency.

The role of static covariates in the charging profile prediction model is examined through a series of feature ablation studies, with results shown in Fig. 6d. These experiments assess the model's performance when trained with all five static covariates compared to versions in which specific covariates are removed. The findings indicate that removing either estimated battery capacity or ambient temperature slightly degrades predictive accuracy, particularly when the model is given limited time series input. The effect is more pronounced for estimated capacity, which provides crucial contextual information in the early stages of a charging session. However, as more temporal data become available, the performance differences between the full and ablated models diminish considerably. This suggests that static covariates are especially valuable for early predictions, compensating for the absence of sufficient time-dependent structure, but their marginal contribution decreases as the charging profile becomes more fully observed. Additional methodological details regarding these ablation studies are provided in Supplementary Fig. 6. To better understand how the model internally prioritises these inputs, the relative importance of the five static covariates was examined using the variable selection networks (VSNs) embedded within the TFT architecture. These networks dynamically assess and weight static features at inference time, suppressing irrelevant or redundant inputs while amplifying those most predictive for a given session. Figure 6e shows the distribution of relative importance scores across the

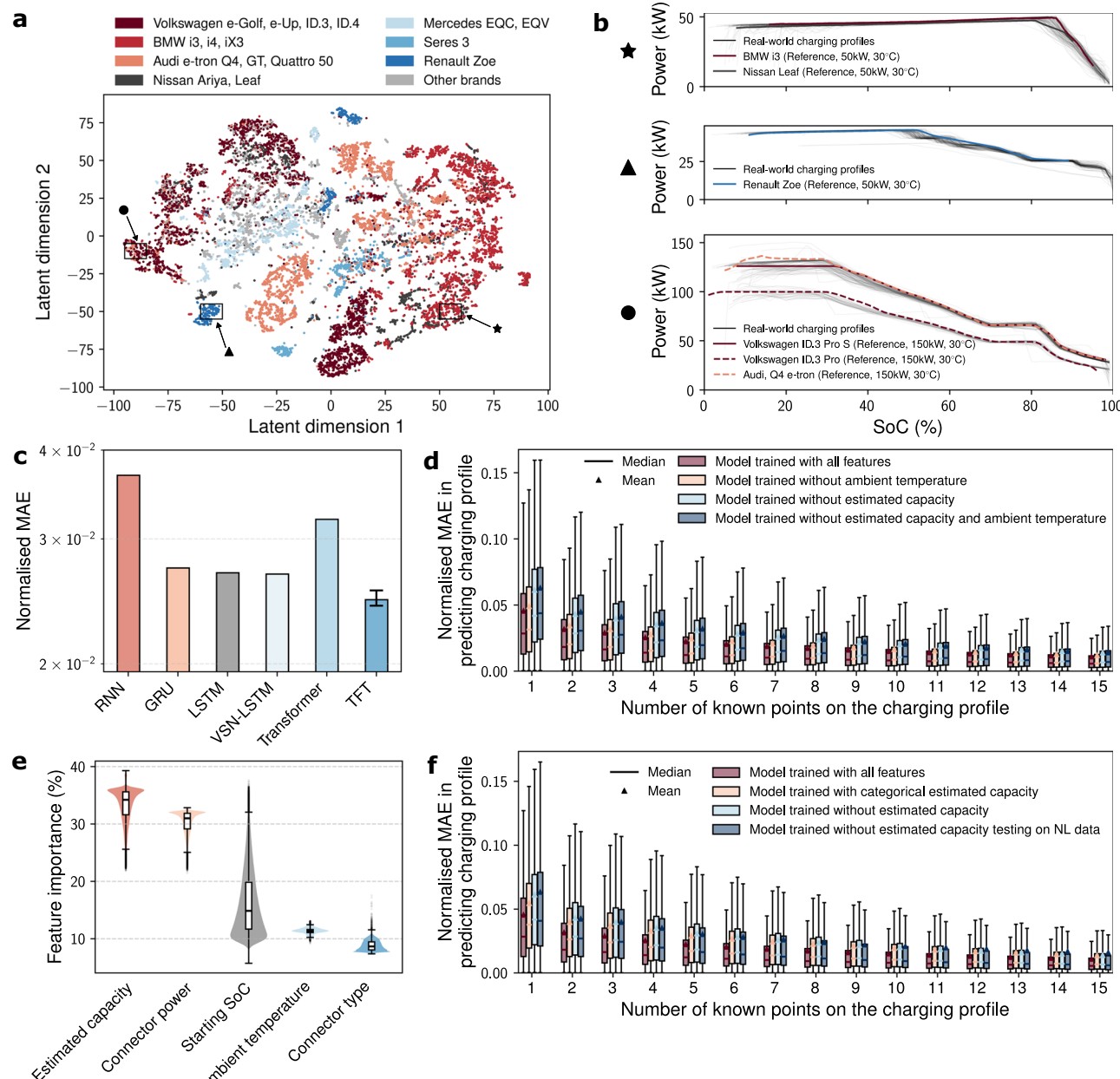

**Fig. 6 | Interpretations of the learned representations and ablation studies of the charging profile prediction model.** Box plots show the median (centre line), interquartile range (box limits), and whiskers (1.5 × the interquartile range). Results integrate data from both test sets. **a** t-distributed stochastic neighbour embedding (t-SNE) visualisation of the latent representations of charging data. Each point represents a full power/state-of-charge (SoC) charging profile, coloured according to its closest match with a reference profile obtained through charging under ideal laboratory conditions. **b** Zoomed-in views of the clusters formed in the t-SNE plot. The black curves, rendered with partial transparency, denote multiple charging profiles, and their overlap yields a grey appearance through cumulative opacity. **c** Performance of the temporal fusion transformer (TFT)-based charging profile prediction model compared against various baselines when evaluated on a combined dataset of test sets one and two. For TFT, three runs are shown directly as the lower error bar, bar height, and upper error bar; other models show a single run. **d** Performance of the charging profile prediction model trained with different configurations of static covariates, evaluated with normalised mean absolute error (MAE) in prediction. **e** Relative importance of the static covariates derived from the charging profile prediction model's variable selection networks. The violin plots represent scaled kernel density estimates of relative feature importance, with overlaid box plots indicating medians and interquartile ranges for each static covariate. **f** Performance of the charging profile prediction model trained with three configurations of battery capacity: omitted, used as a continuous covariate, and used as a binned categorical covariate. For geographical generalisation, the capacity-omitted model is also evaluated on the Netherlands dataset, which lacks this feature.

dataset. All five covariates are found to contribute meaningfully to the prediction task. On average, estimated capacity emerges as the most influential static feature, followed by connector power rating. The importance of starting SoC shows considerable variability across sessions, with higher weights typically assigned to more extreme starting SoC values. In contrast, ambient temperature and connector type generally play less prominent roles, though their influence remains non-trivial. It is important to note that these relative importance values reflect the internal priorities of the model rather than an intrinsic ranking of physical relevance. Covariates with lower importance scores may have effects that are more predictable, more subtle, or captured indirectly through correlations with time series features. Moreover,

the importance distributions are specific to the structure and distribution of the current dataset and may vary under different conditions or in other deployment scenarios.

To assess the model's ability to generalise geographically, an evaluation was conducted on another held-out test set comprising charging sessions only from the Netherlands, which was not included in training. As this dataset lacked battery capacity information, the model trained without estimated capacity was used. As shown in Fig. 6f, this model achieved comparable performance on the Netherlands data and the primary test sets from the UK and Germany, indicating a degree of robustness to geographic distribution shift within Western Europe, even under missing covariates. Figure 6f also presents an ablation study examining the effect of reducing the granularity of the estimated capacity input. Rather than using continuous values, estimated battery capacity was discretised into four broad categories (10 to 50, 50 to 80, 80 to 120, and more than 120 kWh). This binned representation yielded intermediate performance, outperforming the model trained without capacity but underperforming relative to the continuous variant. These performance differences suggest that estimated capacity conveys more than nominal pack size, it may implicitly encode additional battery-specific characteristics, such as cell chemistry or charging rate limitations. Given that it is the only static input varying systematically across EVs, while other inputs primarily reflect session-specific conditions, this covariate likely acts as a proxy for EV identity, enabling the model to infer latent structural or electrochemical differences. While such associations cannot be directly verified from the available data, the observed performance degradation upon discretisation supports the view that preserving the continuous representation allows the model to exploit fine-grained distinctions relevant to real-world charging behaviour.

## Discussion

The growing prevalence of EVs and the increasing demand for rapid, reliable public charging infrastructure necessitate advanced predictive tools to accurately model charging patterns and durations, in order to enhance user experience, to facilitate the efficient management and planning of charging systems, and reduce the impact of EV charging on the grid. This work introduces a deep learning framework that predicts DC fast charging profiles at scale across multiple EV models, a range of charger power ratings, and diverse real-world contexts. The model is trained on a dataset of 909,135 real-world DCFC charging sessions from the UK and Germany, achieving 95% accuracy with an absolute error of less than one minute when predicting charging durations using six data points (power as a function of SoC) collected within five minutes. These results demonstrate the model's robustness and its ability to generalise across different EV models and charging scenarios.

The model is designed for practical deployment. It operates in an event-driven fashion, updating predictions as new SoC readings are received, typically every one minute. Inference is computationally efficient: a single forward pass over 1024 sessions completes in approximately 100 milliseconds on standard GPU hardware, as detailed in Supplementary Table 1, corresponding to a throughput of over 10,000 sessions per second. This is several orders of magnitude faster than the rate at which new input data becomes available, confirming that the method is capable of real-time operation even at scale.

Crucially, the model relies solely on standard charging session data available at the charger, namely charging power, SoC, and charger metadata. For optimal performance, additional covariates such as estimated battery capacity and ambient temperature should be used when available, and can be inferred or measured locally during the charging session. However, the model remains capable of delivering robust predictions in their absence, maintaining usability in settings where such information is incomplete or unavailable. This design supports broad applicability across diverse EV and infrastructure configurations.

The proposed method enables a range of applications across both system-level and consumer-facing domains. For distribution and transmission system operators (DSOs and TSOs), accurate near-term forecasts of charging behaviour based on minimal early-session data can facilitate more precise demand prediction and flexible load management. This is particularly relevant for grid-constrained areas with high EV adoption, where real-time profiling at scale can support congestion mitigation, dynamic tariff design, and anticipatory grid services such as demand shifting or curtailment strategies. On the consumer side, the method provides the technical foundation for features such as real-time charging duration estimates, adaptive pricing notifications, and enhanced queueing logic at high-traffic charging sites. These use cases benefit from the model's ability to operate on a session-by-session basis without relying on persistent vehicle identifiers or proprietary internal vehicle data.

While the charging profile prediction model has been shown to be able to generalise to session data collected in the Netherlands, which is a geographically distinct region from the training data, also within northwestern Europe, full adaptation to radically different markets or climate zones may require additional training data. However, the model architecture and self-supervised learning workflow are sufficiently flexible to accommodate such extensions. Future work could include cross-session user modelling, personalised prediction, or behavioural inference. However, such directions would raise new privacy considerations, particularly if persistent identifiers or richer metadata are introduced. In these cases, privacy-preserving techniques such as differential privacy or federated learning may be necessary to ensure appropriate handling of both user-level data and commercially sensitive infrastructure information. Further gains may be realised through model-based optimisation, informing real-time decisions such as charger assignment, load scheduling, or dynamic pricing based on predicted charging behaviour. The charging profile prediction model may also serve as a forward simulator within reinforcement learning frameworks for smart charging control. More advanced uncertainty quantification techniques could further improve transparency in safety or cost-sensitive applications.

## Methods
### Detailed model architectures

In both the anomaly detection and the charging profile prediction model, the feature projection layer includes linear projection layers for numeric time series including power and SoC, and the static covariates of: starting SoC, connector type (i.e., CHAdeMO or Type 2 CCS), connector power, estimated battery capacity, and ambient temperature. A trainable embedding layer was used for the connector type, which is a categorical static covariate. The hidden state sizes of all subsequent layers were aligned with the size of the projection layer to maintain consistency. The static covariates were utilised to augment the time series variables in several ways, including as contexts in variable selection networks (VSNs) and gated residual networks (GRNs), as well as to initialise the cell and hidden states in the LSTM layers.

### Anomaly detection model

The $\beta$-VAE-based anomaly detection model compresses the charging profiles with lengths of up to 101, to a much smaller latent space through its encoder, which captures the most essential features of the data. Its decoder then reconstructs the charging profiles from this latent representation. During training, the model learns to minimise the difference between the original charging profile and its reconstruction, as well as to regularise the latent space distribution to follow that of a standard Gaussian. Anomalies are identified by their high reconstruction error, as the model struggles to accurately reconstruct inputs that deviate significantly from the patterns seen in the majority of the data.

The anomaly detection model was trained on 810,765 charging sessions and validated on 90,086 sessions. Post-training, 9334 sessions – the 1% of the dataset with the highest reconstruction errors – as well as additional sessions where the computed charging time of the reconstructed profile differed from the ground truth by more than 15 minutes were intentionally excluded from further use to prevent potential outlier influence.

The anomaly detection model was trained to jointly minimise the reconstruction loss and the KL divergence loss. The total loss for a single charging session $\mathbf{y}$ is defined as:

$$\mathcal{L}(\mathbf{W}; \mathbf{y}) = \mathcal{L}_{\text{recon}}(\mathbf{W}; \mathbf{y}) + \beta \cdot \mathcal{L}_{\text{KL}}(\mathbf{W}; \mathbf{y}), \tag{1}$$

where $\mathbf{W}$ denotes the weights of the model parameters and $\beta$ controls the degree to which the KL divergence term contributes to the total loss function.

The mean squared error (MSE) based reconstruction loss $\mathcal{L}_{\text{recon}}$ and the KL divergence loss are computed as:

$$\mathcal{L}_{\text{recon}}(\mathbf{W}; \mathbf{y}) = \frac{1}{\sum_j m_{\text{pad},j}} \| \mathbf{m}_{\text{pad}} \odot (\mathbf{y} - \hat{\mathbf{y}}) \|^2,$$
$$\mathcal{L}_{\text{KL}}(\mathbf{W}; \mathbf{y}) = \frac{1}{2} \sum_{k=1}^{K} (\mu_k^2 + \sigma_k^2 - 1 - \log(\sigma_k^2)), \tag{2}$$

where $\mathbf{m}_{\text{pad}}$ is a boolean masking vector used to differentiate between true and padded positions in the charging profiles, $\hat{\mathbf{y}}$ represents the reconstructed charging profiles, and the operator $\odot$ denotes the Hadamard product. $K$ is the dimensionality of the latent space, $\mu_k$ is the mean and $\sigma_k$ is the standard deviation of the latent variables encoded from $\mathbf{y}$.

A beta-annealing strategy was used to prioritise training the encoder in early epochs of the training phase and prevent posterior collapse. The parameter $\beta$ was dynamically adjusted at each training epoch $t$ according to the following logistic schedule:

$$\beta = \frac{1}{1 + \exp(-k \cdot (t - t_0))} \cdot \beta_0, \tag{3}$$

where $k$ is a constant that controls the steepness of the logistic function, set to 0.1, and $t_0$ is the epoch at which $\beta$ reaches half of its maximum value $\beta_0$. For this work, $t_0$ was set to 90 with $\beta_0$ fixed at $10^{-5}$. The anomaly detection model was configured with a state size of 128 for all its layers; both the LSTM encoder and decoder had three LSTM layers, and the model in total had 1.5 million trainable parameters. Through experimentation, it was determined that the dataset could be effectively compressed to a latent size of six without a significant impact on the reconstruction loss. The model training converged in about 250 epochs with a batch size of 512 and a maximum learning rate of $10^{-3}$ using the AdamW optimiser[42]. The training process took about 40 NVIDIA A100 GPU hours.

**Charging profile prediction model**
The charging profile model predicts complete power-SoC charging profiles by estimating the quantiles of the power and the associated charging times, using a partial charging profile and a set of static covariates as inputs. The partial profile can consist of as little as a single data point. During training, a random proportion, of size ranging from 10% to 99%, of the charging profile is deliberately masked to enable the model to learn to reconstruct the full profile from the remaining partial data. The masking predominantly targets the latter portion of the profile but can also be applied to intermediate segments. During testing, only the rear portions of the charging profiles are masked. The model architecture is constructed to operate causally, meaning that it makes predictions based solely on past and present data, without relying on information from future points on the charging profile.

Following the anomaly detection step, the charging profile prediction model was trained on 713,213 sessions collected between November 2021 and 16 June 2024. Model performance was evaluated on two distinct test sets, with test set one comprising 44,575 sessions drawn from the same time frame as the training data, and test set two with 44,578 sessions collected from 17 June to 9 July 2024.

The charging profile prediction model was optimised to minimise the total quantile loss, with the quantile loss for a single charging session $\mathbf{y}$ defined as:

$$\mathcal{L}_{\text{quantile}}(\mathbf{W}; \mathbf{y}) = \sum_q \mathcal{L}_q(\mathbf{W}; \mathbf{y}), \tag{4}$$

where $\mathbf{W}$ denoted the weights of the model parameters, $q$ is the desired quantile with $q \in \{0.1, 0.25, 0.5, 0.75, 0.9\}$, and $\mathcal{L}_q$ is the quantile-specific loss, expressed as:

$$\mathcal{L}_q(\mathbf{W}; \mathbf{y}) = \frac{1}{\sum_j m_j} \mathbf{m} \odot \left[ q(\mathbf{y} - \hat{\mathbf{y}}_q)_+ + (1 - q)(\hat{\mathbf{y}}_q - \mathbf{y})_+ \right], \tag{5}$$

where $\hat{\mathbf{y}}_q$ is the predicted charging profile for quantile $q$, and $(\cdot)_+ = \max(0, \cdot)$. The masking term $\mathbf{m}$ is defined as:

$$\mathbf{m} = \mathbf{m}_{\text{pad}} \odot (\mathbf{m}_{\text{hidden}} + \alpha \mathbf{m}_{\text{train}}), \tag{6}$$

where the boolean vector $\mathbf{m}_{\text{pad}}$ differentiates real from padded positions, $\mathbf{m}_{\text{hidden}}$ identifies positions within the predicted curve, and $\mathbf{m}_{\text{train}}$ corresponds to the positions used as input for the charging profile. The scaling factor $\alpha$ is set to 0.1.

The model architecture includes five LSTM layers and three multi-head attention layers, each with four attention heads, resulting in a total of 9.8 million trainable parameters. All layers have a state size of 256. Training of the charging profile prediction model converged in approximately 350 epochs with a batch size of 1024, using the AdamW optimiser and a maximum learning rate of $5 \times 10^{-4}$. The training process took about 72 NVIDIA A100 GPUs hours.

**Charging duration computation**
Session charging duration can be computed from the charging profile as follows:

$$T_{\text{session}} = \int_{s_{\text{start}}}^{s_{\text{stop}}} \frac{\hat{c}}{p(s)} \, ds, \tag{7}$$

where $T_{\text{session}}$ is the session charging duration in hours, $s_{\text{start}}$ and $s_{\text{stop}}$ are the starting and stopping SoC respectively, $\hat{c}$ is the estimated capacity of the EV in kWh and $p(s)$ is the power in kW as a function of the SoC $s$.

## Data availability
The dataset used in this study contains proprietary information and is subject to a Non-Disclosure Agreement (NDA) with Shell. Access is therefore restricted and can only be granted upon approval of an NDA. Researchers interested in obtaining the data may contact R.D. to request access. The dataset will be made available solely for research purposes under the terms of this agreement and in compliance with applicable data protection regulations. Source data are provided with this paper.

## Code availability
The code used in this study is available at https://github.com/acse-sl420/ev_charging_ml. An archived snapshot corresponding to the

published version is deposited on Zenodo under the https://doi.org/10.5281/zenodo.17183022. The pretrained models generated in this work are deposited separately on Zenodo under the https://doi.org/10.5281/zenodo.17183746.

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

## Acknowledgements
The authors are grateful to Simon C. Warder, Balakrishnan Bhaskaran, and Xin Zhong for meaningful discussions that contributed to the development of this work, and to Francois van Schalkwyk for facilitating the use of computational resources. S.L., M.Z., and M.D.P. acknowledge Shell Research Limited for funding this work.

## Author contributions
S.L. conceived the study, processed the dataset, implemented the algorithms, conducted the numerical experiments, generated the figures, and wrote the manuscript. M.Z. contributed to performing the numerical experiments. R.D. provided access to the dataset, offered business context, contributed to interpreting the results, and reviewed the manuscript. B.R. contributed to the business context and result interpretation. M.D.P. conceived the study, contributed to result interpretation, reviewed the manuscript, and supervised the project.

## Competing interests
The authors declare no competing interests.
