## [Transparent Peer Review file · Nature Communications]

Artificial intelligence predicts real-world EV DC charging profiles and durations

Corresponding Author: Mr Siyi Li

Version 0:

Reviewer comments:

Reviewer #1

(Remarks to the Author)

The manuscript, "Artificial intelligence predicts real-world EV DC charging profiles with minimal inputs," introduces an innovative deep learning framework leveraging a large dataset of 900,830 real-world DCFC sessions. Overall, the manuscript is well-organized, features high-quality and visually appealing figures, and addresses a significant challenge in electric vehicle (EV) charging prediction. The approach, which uses minimal input parameters to predict charging profiles and durations with real-time updates and uncertainty estimation, holds substantial potential to improve EV charging infrastructure and user experience. While the paper is suitable for publication, several major technical concerns should be comprehensively addressed through a major revision.

Major Comments:

1. Clarify the definition of "minimal inputs." Specifically, quantify and justify what constitutes sufficient data for accurate predictions. Discuss the anticipated performance relative to varying data availability.
2. The paper claims 90% accuracy from a single data point. Authors should clearly justify why this accuracy level is acceptable or sufficient for practical EV applications.
3. Validate thoroughly that the achieved "90% accuracy from one data point" is not due to overfitting. Conduct multiple rounds of random shuffling and cross-validation of the dataset to confirm the robustness of results.
4. Examine carefully the claim regarding "unseen data." Simply excluding data from training doesn't necessarily mean data are truly "unseen" if their distribution closely mirrors training data. Provide distribution analyses and assess performance explicitly on out-of-distribution scenarios.
5. The assertion "current predictive methods are often not adapted for real-world conditions nor validated with comprehensive datasets" is questionable. The authors should carefully review and reference recent works that have extensively handled random, scarce, and heterogeneous data with machine learning (e.g., Journal of Energy Storage, vol. 49, p. 104132, May 2022, doi: 10.1016/j.est.2022.104132, Nat Commun 15, 10154 (2024), Nat Commun 14, 8032 (2023), ACS Energy Lett. 2023, 8, 8, 3269–3279, Journal of Power Sources Volume 597, 30 March 2024, 234156) and revise this claim accordingly to avoid misrepresenting the current state of research.
6. Given the claim of real-time usability, the authors should validate their method using commercial BMS hardware parameters and include clear computational latency metrics relevant for real-time deployment.
7. Although the dataset is under NDA, reviewers should have access to the data and code confidentially for validation purposes in the next revision round. Public release remains at the authors' discretion.
8. Better articulate the specific research gaps being addressed. Although IP and privacy are mentioned as concerns, the manuscript lacks clear methods related to privacy protection, such as differential privacy or federated learning. Please clarify this connection or revise the discussion accordingly.
9. Clarify the logical relationship between the anomaly detection and the charging profile prediction modules to ensure coherence and methodological transparency.
10. The approach of classifying anomalies based solely on reconstruction curve errors and charging time estimates may introduce substantial uncertainty due to the complexity of real-world conditions. Authors need to discuss and justify the robustness and validity of this anomaly classification.
11. Provide detailed analyses regarding potential information leakage or biases that could arise by excluding anomalous data during training.
12. Enhance the evaluation rigor by performing ablation studies. Specifically, replace or remove specific model components to assess their individual contributions and overall model effectiveness.
13. Incorporate comprehensive comparisons with baseline models such as LSTM variants, recent deep learning methods

like other Transformer variants, and traditional time-series forecasting techniques. Such comparisons are essential to convincingly demonstrate the proposed model's superiority and generalizability.

Minor Comments:

1. The model design is interesting, but lacks sufficient detail. It is recommended to provide detailed model parameters and training hyperparameters, supplement with pseudocode or a model flowchart, and elaborate on data preprocessing and feature engineering
2. The manuscript mentions "1% reconstruction error and a 15-minute charging time difference as the anomaly detection threshold." Is there any theoretical basis for this? It is recommended to further discuss the rationality of the threshold.
3. The model's inputs include charging power and battery SoC sequences. How is the SoC in this paper obtained, and how is the accuracy of the SoC ensured? Additionally, does the SoC have any impact on the deep learning method proposed in this paper?
4. Since the dataset used in the manuscript contains proprietary information and is subject to confidentiality agreements, the authors do not disclose the data. Therefore, it is recommended to provide some non-confidential data or supplement the article with a more detailed description of the data to facilitate learning and verification by other researchers.
5. The design of the figures in the manuscript is good, but the text in some figures is too small. It is recommended to ensure that the text is clear at 100% view, in accordance with the journal's requirements.
6. The manuscript mentions that "The grouping for Nissan 315 Ariya/Leaf is less obvious." It is recommended to conduct an in-depth analysis of the reasons for similar situations.
7. Please discuss economic feasibility of the method and uncertainty analysis, as they are crucial for practical deployment conditions.
8. The anomaly detection process filters data based on reconstruction error and charging time discrepancies (>15 minutes), but the rationale behind the threshold selection (e.g., whether it is based on a quantile of the data distribution) is not provided. Clarifying this aspect would improve the model's transparency and reliability.
9. The dataset is limited to Europe's temperate climate (-14°C to 35°C). A discussion on the model's applicability in extreme climates (e.g., tropical or polar regions) or experiments involving cross-region transfer learning would enhance the work's global relevance.
10. While the paper includes a variable importance analysis, a deeper exploration of the physical interpretability of key factors (e.g., battery capacity) would strengthen the analysis and improve the model's transparency.

(Remarks on code availability)

Reviewer #2

(Remarks to the Author)

(Remarks on code availability)

The reviewer can see the code but without the data. Data needs to be seen very clearly.

Reviewer #3

(Remarks to the Author)

This paper presents a deep learning model that predicts EV charging profiles and durations using minimal input from over 900,000 real-world DC fast charging sessions. The model operates in real time, provides uncertainty estimates, and achieves up to 95% accuracy with data collected within the first five minutes of charging. The topic is timely and relevant for practical EV charging applications. Notably, the model demonstrates the generalization across different EV models and delivers high prediction accuracy early in the charging process. Overall, this paper is not only well-organized and is highly valuable for the EV applications.

- In Figure 2a, the proposed framework is shown to dynamically adapt to real-time data using a cloud server and HPC platform. Please provide comments on the computational load and feasibility for practical, large-scale deployment.
- Would battery chemistry be considered an important covariate in the model? A discussion on this aspect would be helpful, particularly considering the variability across different EV models.
- Please elaborate on how the influence of covariates might change as the dataset becomes more diverse, especially in terms of EV types, battery chemistries, and user charging behaviors.

(Remarks on code availability)

Reviewer #4

(Remarks to the Author)

General comments:

The topic of this paper is of high relevance. The performance of the novel method is thoroughly demonstrated and proved here.

The overall structure of the paper is coherent.

In the following, I will elaborate on different aspects that would improve the scientific quality of this manuscript and also the understanding to the reader.

Title:

The title is very strong. I suggest to include also "duration", f.e. "AI predicts real-world EV DC charging profiles and charging duration with minimal inputs"

In Main:

- The state of the art discussion is missing. There reader should get some idea of what the current methods in this field are capable of in terms of the performance parameters that are novel in your approach.

- Why use predictive models and not a physical one? What are the donesides of using the relevant parameters and compute a physical model?

- One of the interesting aspects is the anomaly detection; the authors should introduce this concept in more proper way in the beginning of the text and than later also go into typical applications for forecasting of this and why this is essential to the approach.

comments on the method description:

Mention the temporal resolution.

-This is more of a general comment. But this information should by already included in the abstract, then main, should be also part of the state of the art. There is no mention of the temporal resolution. But this is specifically when it comes to electricity market applications one of the core parameters, specifically for congestion managment or grid stability. The same goes for the prediction of the charging profile? How long does it take to generate this?

Results:

- you argue in 95% of cases, that there is a 1 minute error -> what exactly is meant of this? that the difference between the forecasted power level and between the actual is one minute? Or is this the duration? I would more expect that when charging profiles are predicted, the power level is also of interest. For energy-related applications, the error in power level is definitely more of interest.

Comments on the figures:

Figure 1d:

I suggest to add 'estimated *battery* capacity' for better understanding

Figure 2:

- nice workflow description

- Maybe highlight the step of charging profile/time predictive which is the novelty that you contribute

- also it seems that only a-c are really relevant /d-f do seem more detail

- I suggest to make the plots b) and c) more readable by f.e. make the text horizontal instead of vertical

figure 3:

(b) the three different cases are not understandable from the beginning; i suggest to add a description: example 1/2/3

Discussion:

I expected stronger points to be made in the discussion. Currently, the discussion rather is a summary.

The following points should be addressed:

- What applications can this new minimal-input method with the given performance parameters enhance? The relevancy for dso and tso operation; as well as consumer-side benefits and applications?

- What does this method allow to improve in these applications? (compared to the current state of the art)

You address the application to different geographic regions. Mention here again (and also this is missing and should be included in the Abstract) the region of the charging profiles for training.

Elaborate more on what "different climates, behaviors, infrastructure" are?

Future work should be addressed.

(Remarks on code availability)

Version 1:

Reviewer comments:

Reviewer #1

(Remarks to the Author)

The reviewer appreciated authors' efforts in addressing the comments and the quality of paper has been improved. However, the reviewer finds some issues to be carefully addressed before publication.

Remaining Comments:

1. While the reviewer acknowledges that the proposed model is designed to operate with a small segment of input data, the use of the term "minimal data" in the paper title feels overly strong and somewhat vague. It is recommended to soften this claim to better reflect the practical limitations and context of the approach.
2. The reviewer found that Energy Environ. Sci., 2025,18, 7413-7426 (Immediate remaining capacity estimation of heterogeneous second-life lithium-ion batteries via deep generative transfer learning) has a similar motivation using minimal data to make predictions of battery states using a VAE architecture. Please consider discussing this in the introduction to enrich the depth of this paper.
3. The reported accuracy of ~90% is presented without sufficient contextual justification. Without a clear explanation of the target application's tolerance for error, this number may lack practical significance. It is advised to either clarify why this level of accuracy is acceptable or avoid highlighting it as a standalone metric.
4. To ensure that the reported performance is not the result of overfitting, the authors should consider including a random shuffling experiment or adding training vs. validation loss plots. This would help assess whether the model's high accuracy is generalizable or confined to the specific dataset used.
5. Although the paper states that the test data was completely withheld during training, it is possible that the testing samples lie within the subspace of the training distribution. Since Wasserstein distance alone may not offer intuitive insight into data separation, it would be helpful to include a visualization of the data distributions (e.g., PCA or t-SNE plots) to support the claim that the test set truly represents unseen conditions.

(Remarks on code availability)

Reviewer #2

(Remarks to the Author)

(Remarks on code availability)

The data and code are not fully disclosed due to claimed restrictions.

Reviewer #3

(Remarks to the Author)

Thank you for the detailed responses and clarifications. I recommend accepting this manuscript for publication in the journal.

(Remarks on code availability)

Reviewer #4

(Remarks to the Author)

Dear authors,

I believe that most the the criticism I had has been addressed and the readability as well as scientific quality of the manuscript has been significantly improved.

I have major issue that I have already addressed during the first round of the review - maybe not directly: The visualizations include a lot of information which I still don't find necessary and overall disruptive to the communication of your research. For example: d-f in Figure 2 - These are not discussed within the text. Just the visualization is included. Removing these subfigures could give way better opportunity to explain a-c. The arrangement of the figures is rather confusing and just looking at the visualizations there is no natural visual guidance. I understand the necessity to display the information but I believe that for this interdisciplinary journal, intuitive visualizations are essential.

(Remarks on code availability)

Version 2:

Reviewer comments:

Reviewer #1

(Remarks to the Author)

In the revised manuscript, it is great to see the authors' effort to address of all concerns from reviewers. Good job!

(Remarks on code availability)

Reviewer #2

(Remarks to the Author)

(Remarks on code availability)

Please publish the code and data if the paper will be accepted for publication.

Authors' Response to Reviewer 1

General Comments. The manuscript, "Artificial intelligence predicts real-world EV DC charging profiles with minimal inputs," introduces an innovative deep learning framework leveraging a large dataset of 900,830 real-world DCFC sessions. Overall, the manuscript is well-organized, features high-quality and visually appealing figures, and addresses a significant challenge in electric vehicle (EV) charging prediction. The approach, which uses minimal input parameters to predict charging profiles and durations with real-time updates and uncertainty estimation, holds substantial potential to improve EV charging infrastructure and user experience. While the paper is suitable for publication, several major technical concerns should be comprehensively addressed through a major revision.

Author response to general comment: Thank you for your thoughtful and constructive review. We sincerely appreciate your positive assessment of the manuscript and your recognition that it is well-organized, features high-quality and visually appealing figures, and addresses a significant challenge in electric vehicle charging prediction. We are especially encouraged by your view that the proposed approach holds substantial potential to improve EV charging infrastructure and user experience.

We also thank you for your detailed feedback and insightful technical comments. These comments have been extremely valuable in guiding our revisions, and we believe that we have addressed all of your comments in detail below.

Major Comment 1

Clarify the definition of "minimal inputs." Specifically, quantify and justify what constitutes sufficient data for accurate predictions. Discuss the anticipated performance relative to varying data availability.

Author response to major comment 1: Thank you for your comment.

The input to the proposed charging profile prediction model consists of two parts: a partial Power/state-of-charge charging profile, which can be as minimal as a single point, and optionally, a series of static covariates, that include starting state of charge (SoC), estimated battery capacity, ambient temperature, charger power and charger type. Starting SoC, charger power and charger type are always known to the charger, whereas an estimate for battery capacity can be computed as charging progresses, and ambient temperature can be measured directly. As we've shown in the feature ablation studies in Figure 5 of the manuscript, these static covariates improves prediction accuracy, but the model can still deliver robust predictions without them.

The absolute minimum amount of data that can be used to generate a prediction by the charging profile prediction model is a single data point on the Power/SoC charging profile, this will typically be the first data point reading generated by the charger. This allows customers to receive a preliminary prediction almost immediately after plugging in. There is no minimum duration required to capture a sufficient portion of the charging profile, rather, the accuracy and certainty of the prediction naturally improve over time as more data become available. We showed in panel a of Figure 1a (Figure 4 in the revised manuscript) that when a single point is known on the charging profile, the charging profile model reported about 0.045 normalised MAE in predicting the charging

profiles, and 10% relative error in predicting charging duration or an average absolute error of 2 minutes. We showed in panel b that this corresponded to an R-squared value of 0.92 and in panel c that, for more than 80% of the sessions in the test sets, this prediction is more than 85% accurate.

The model predictions does not need to stop there, as the proposed model is trained with self-supervised learning and can take as input time series of varying lengths, the model can continue to make predictions and refine them as the charging progresses. We report the distribution of errors in the predictions as the charging progresses and increasing number of known points on the charging profile are reported to the charger and used as input to the model in the box plots of Figure 1a. It could be clearly seen that as the session progresses and more SoC–power pairs are incorporated, the model’s predictions systematically improve, with the distribution of errors narrowing and converging toward higher accuracy.

We note that definitions of “minimal data” or thresholds for “sufficient accuracy” are inherently subjective and context-dependent. Accordingly, we provide a range of accuracy metrics across input sizes to allow readers to make their own assessment based on the operational requirements most relevant to their applications.

Figure 1: Performance evaluation of the charging profile prediction model, where the model’s median predictions were compared to ground truth charging profiles and times. Charging times were calculated based on either the completion of the charging session or reaching 80% SoC, whichever came first. Analysis was confined to sessions with a strictly greater than 15% change in SoC. **a** The charging profile prediction model’s performance on predicting charging profile and charging time for different numbers of known points of the charging profile as input, evaluated on the two test sets. Normalised mean absolute error (MAE) was computed as the MAE of predicted power normalised by connector power rating. The error bars represent the interquartile range (IQR). **b** Scatter plot with density overlays comparing median predicted charging times against ground truth across different number of input points. The results combine data from both test sets. **c** The accuracy ranges of the test set sessions, evaluated on both test sets. **d** The percentage number of uncertain sessions for different number of input points across both test sets. **e** The median time required to accumulate a specific number of points on the charging profile, with error bars representing the IQR. **f** Distribution analysis of relative and absolute errors in predicting charging time, across varying numbers of input points on the charging profile, incorporating results from both test sets.

Major Comment 2

The paper claims 90% accuracy from a single data point. Authors should clearly justify why this accuracy level is acceptable or sufficient for practical EV applications.

Author response to major comment 2: Thank you for your comment and for raising this point and we welcome the opportunity to clarify. This particular reported result, i.e. that the charging profile prediction model achieves approximately 90% accuracy on the held-out tests from a single Power-SoC observation, is *not* presented as a stand-alone claim of sufficiency for deployment, but rather as a transparent and informative benchmark. Our intention is not to suggest that one should operate the model from a single point alone (which one could), but to show that even in such data sparse conditions, the model remains capable of producing meaningful and usable predictions, which is something that most existing approaches are unable to do, as they often require a minimal charging trajectory of certain lengths or fixed-length time series inputs. From the customer’s perspective, this capability has immediate practical value: they receive a useful prediction shortly after plugging in, without having to wait for an extended portion of the charging process to elapse for an estimate.

Importantly, our framework is designed to operate in an event-driven, incremental fashion: predictions are made continuously as new data points arrive, and the model updates its estimates accordingly. Because it is trained using self-supervised objectives on variable-length sequences, it is fully capable of refining predictions as the session progresses. This design enables prediction at a single input point on the power-SoC charging profile, two points, three points, and so on, with performance improving as more information becomes available, which is an ability illustrated throughout Figure 4 in the revised manuscript. The result reported at a single data point simply reflects the lower bound of this capability.

We include these early-stage metrics not as a statement of sufficiency, but for completeness and transparency, to demonstrate the model’s behaviour under minimal data conditions. We defer to practitioners to determine what level of accuracy is acceptable for a given application. Our goal is to equip the readers with a detailed understanding of the model’s capabilities across a range of operating regimes.

Major Comment 3

Validate thoroughly that the achieved "90% accuracy from one data point" is not due to overfitting. Conduct multiple rounds of random shuffling and cross-validation of the dataset to confirm the robustness of results.

Author response to major comment 3: Thank you for your comment. To assess robustness and mitigate concerns about overfitting, we conducted three independent training runs using different random seeds and data splits. The results reported in Figure 4 of the revised manuscript are based on a model averaged ensemble across these three runs. While additional rounds of cross-validation or k-fold cross-validation could further characterise performance variability, they were not pursued due to the substantial computational cost—each training run requiring approximately 72 A100 GPU hours.

We note that the top-performing individual model slightly outperformed the ensemble. However, we chose to report ensemble results for transparency and because they offer a more stable and representative estimate of

real-world performance across varying initialisations and data partitions.

We have added the following text to the subsection “Time to reliable prediction operationally post-plug-in”:

The performance of the charging profile prediction model is detailed in Figure 4. All results reported correspond to an ensemble model constructed by taking a weighted mean of predictions from three independently trained instances of the TFT model, each initialised with different random seeds and data splits. Although the best-performing individual model achieved slightly higher accuracy on certain metrics, the ensemble was selected for analysis as it provides greater robustness and is more representative of typical model behaviour under varied conditions.

Major Comment 4

Examine carefully the claim regarding "unseen data." Simply excluding data from training doesn't necessarily mean data are truly "unseen" if their distribution closely mirrors training data. Provide distribution analyses and assess performance explicitly on out-of-distribution scenarios.

Author response to major comment 4: Thank you for your comment and for raising this very important point. We agree that assessing generalisation beyond the training distribution is an important consideration. We would like to clarify that, by “unseen” data, we mean sessions that were completely withheld from both the model training stage and the hyperparameter tuning stage, in line with standard machine learning practice.

Given that formal two-sample tests that compare entire distributions are not currently available for variable-length, ragged multivariate time series sequences such as EV charging sessions presented in the dataset, where each session differs in static covariates, time series features and also session lengths, we decided to use the trained β -VAE model to project the multi-variate time series of varying lengths onto a fixed-sized latent space, and compare the distribution shift between the training set and two test sets using the 1-Wasserstein distance metric.

We have made the following edit to the revised manuscript:

Following this anomaly exclusion step, the charging profile prediction model was trained on the filtered dataset and subsequently evaluated on two independent, previously unseen test sets. The first test set consists of contemporaneous sessions collected within the same time frame as the training data, whereas the second test set comprises sessions collected immediately following the period over which the training data was collected, and was designed to simulate real-world operational conditions and assess the model's predictive performance in a near-future context. Distributional differences between the training and test sets are quantified and visualised in Supplementary Figure 2 using latent representations from the β -VAE model.

We added the following to Supplementary Information:

To assess whether the held-out test sets for the charging profile prediction model differ meaningfully from the training data in distribution, a permutation-based statistical test was conducted using the 1-Wasserstein distance as a distributional divergence metric.

For each dataset pair, the observed 1-Wasserstein distance was computed between the training and test sets in the latent space of the trained β -VAE model. Distances were first calculated independently for

each latent dimension, and the average across all dimensions was used as the overall test statistic. The 1-Wasserstein distance quantifies the minimum effort required to morph one probability distribution into another and serves as an interpretable proxy for distributional similarity. Notably, the β -VAE was trained on a random train/test split that differs from the one used for the main charging profile prediction model, ensuring independence between these processes. To evaluate statistical significance, a permutation test with 1,000 iterations was performed under the null hypothesis that both datasets are drawn from the same distribution. In each iteration, all latent vectors from the training and test sets were combined and randomly partitioned into two groups of equal size. The mean 1-Wasserstein distance was computed for each pair, forming a null distribution of distances expected under random assignment. The empirical p -value was calculated as the proportion of permuted distances greater than or equal to the observed distance. The results are illustrated in Figure 2.

Figure 2: **a** 1-Wasserstein distance between train and test sets shows higher observed values than the null, especially for temporal splits ($p < 0.001$). **b** Effect size measured with z-score is minimal for test set one with random split but large for test set two with temporal splits.

While the permutation-based 1-Wasserstein test identified a statistically significant difference between the training set and test set one (random split), the observed effect size is modest ($z = 3.1$). Given the large sample sizes involved, with over 700,000 sessions in the training set and 45,000 in each test set, even minimal variation between distributions can yield low p -values due to high statistical power. It is also possible that projection onto the latent space amplified the subtle differences in distributions. By contrast, the temporal split (test set two) exhibits a much more pronounced deviation from the training distribution ($z = 154.5$, $p < 0.001$), consistent with a genuine distributional shift, as these sessions were drawn from a later time period not covered during training.

That being said, the proposed model is designed to function reliably within the operational domain captured by the training data, which is close to one million EV-charging sessions collected across DC chargers in the UK and Germany, which together reflect the dominant real-world usage patterns in those markets. As is typical for modern machine-learning systems, the proposed predictive model is not intended to extrapolate to entirely novel contexts (e.g., completely new battery chemistries, completely different charging standards, or user behaviours absent from the training data). In a real-world deployment scenario, the emergence of such distributional shifts would naturally lead to performance degradation until those new behaviours are incorporated into the training

set through periodic retraining. At that point, the system is re-aligned with the evolving operational domain. The proposed model, therefore, is designed not to extrapolate beyond its training distribution, but to adapt over time as new data become available. This paradigm reflects the standard practice in applied machine learning and is consistent with the deployment strategies of many data-driven forecasting systems.

The two held-out test sets provide (i) sessions kept unseen throughout model development and (ii) the modest temporal drift encountered in routine operation. Severely out-of-distribution sessions are the ones identified by the anomaly detection model, and are excluded from training and evaluation by design.

Major Comment 5

The assertion “current predictive methods are often not adapted for real-world conditions nor validated with comprehensive datasets” is questionable. The authors should carefully review and reference recent works that have extensively handled random, scarce, and heterogeneous data with machine learning (e.g., Journal of Energy Storage, vol. 49, p. 104132, May 2022, doi: 10.1016/j.est.2022.104132, Nat Commun 15, 10154 (2024), Nat Commun 14, 8032 (2023), ACS Energy Lett. 2023, 8, 8, 3269–3279, Journal of Power Sources Volume 597, 30 March 2024, 234156) and revise this claim accordingly to avoid misrepresenting the current state of research.

Author response to major comment 5: Thank you for your comment and for recommending these excellent and highly relevant articles from the recent literature; we have indeed decided to cite them in the revised manuscript.

The study in [1] shares a similar high-level objective but is comparatively smaller in scale and focused on more controlled conditions. The works of [2, 3, 4, 5] represent substantial advancements in applying machine learning to random, scarce, and heterogeneous battery datasets, with demonstrated success across diverse tasks such as lifetime prediction, battery recycling, and state of health estimation. While these works do not explicitly address charging profile prediction from real-world public fast charging data, they offer important methodological insights, and we view our study as complementary to these efforts by extending predictive ML methods to a new application domain and operational setting.

We have edited the introduction to reflect these changes:

Recent advances in data-driven modelling of lithium-ion battery systems have increasingly addressed challenges associated with random, scarce, and heterogeneous data using sophisticated machine learning techniques [2, 5, 4, 3]. These studies, often based on real-world datasets, have achieved strong performance in tasks such as state of health estimation, lifetime prediction, and retired battery sorting, and represent meaningful progress toward the development of robust and generalisable battery analytics. By contrast, research specifically focused on predicting EV charging profiles remains comparatively limited. Existing work in this area has largely relied on relatively small datasets, often collected under laboratory or semi-controlled conditions using instrumented cells or proprietary battery management system (BMS) data [6, 7, 1, 8, 9]. Such data are frequently tied to specific vehicles and may not be publicly accessible due to design constraints and privacy concerns specific to original equipment manufacturers (OEMs), which limits the broader applicability and scalability of

these approaches. Moreover, while the use of real-world EV data is becoming more common, it has predominantly been applied to tasks such as state-of-health estimation, fault detection, or lifetime prediction [10, 11, 12, 13, 14], rather than to the forward-looking task of charging profile prediction, despite its relevance for smart charging, range forecasting, charger utilisation planning, and energy services.

The line “*current predictive methods are often not adapted for real-world conditions nor validated with comprehensive datasets*” has been removed from the abstract. We only really meant that for predictive methods that specifically target EV charging profile prediction.

Major Comment 6

Given the claim of real-time usability, the authors should validate their method using commercial BMS hardware parameters and include clear computational latency metrics relevant for real-time deployment.

Author response to major comment 6: Thank you for raising the important point of validating real-time applicability. We would like to clarify that this study is intended as a large-scale proof-of-concept, demonstrating the feasibility and accuracy of DC charging profile prediction using real-world charger-side data. Operational integration with commercial DC charging stations would indeed require further engineering and coordination at the system level, which lies beyond the scope of this work.

However, we emphasise that a core strength of our method is that it operates entirely independently of BMS data or parameters or vehicle-side telemetry. It requires only data passively collected by the charger—specifically, timestamped SoC and power readings—which are readily available in public charging infrastructure. This deliberate design choice supports broader applicability and avoids reliance on proprietary or hardware-specific information that is typically unavailable in real-world deployments.

Regarding computational latency, we benchmarked inference times across a range of batch sizes, and summarised the results in Table 3, also in Supplementary Table S-1 in the main text. Even at the largest batch size (1024), a single forward pass completes in approximately 100 milliseconds using standard GPU hardware. Given that the model updates upon receiving new SoC data (typically every one minute) this latency is well within the requirements for real-time responsiveness. The method is highly scalable and capable of handling thousands of sessions per second, making it suitable for deployment in backend systems supporting public charging infrastructure.

We have added the following to Supplementary information in the main text:

Inference benchmarks on the charging profile prediction model are conducted on a dedicated compute node configured with an AMD EPYC 7742 64-core processor and a single 40 GB NVIDIA A100 GPU. The benchmarking evaluates the inference efficiency of the model under varying batch sizes, comparing CPU-only and GPU-accelerated execution timing, with the timing statistics reported over 50 runs. The results are displayed in Table 3.

Table 1: Forward pass timing and throughput across batch sizes on CPU and GPU.

Batch Size	CPU Time (ms)	GPU Time (ms)	CPU Throughput (samples/sec)	GPU Throughput (samples/sec)
1	37.17 ± 0.46	12.13 ± 0.04	26.9	82.4
16	309.91 ± 3.83	12.72 ± 0.02	51.6	1257.5
32	625.27 ± 28.35	13.24 ± 0.06	51.2	2416.2
64	1302.49 ± 42.44	15.39 ± 0.04	49.1	4158.0
128	2759.00 ± 55.99	20.36 ± 1.58	46.4	6287.4
256	5596.13 ± 82.89	32.63 ± 0.11	45.7	7844.4
512	12126.96 ± 130.86	54.69 ± 0.20	42.2	9361.8
1024	24577.94 ± 117.24	101.08 ± 0.26	41.7	10130.7

Major Comment 7

Although the dataset is under NDA, reviewers should have access to the data and code confidentially for validation purposes in the next revision round. Public release remains at the authors’ discretion.

Author response to major comment 7: Thank you for your comment. Shell UK has agreed to provide reviewers access to the dataset under strict confidentiality. The full anonymised dataset and pre-trained models are now included in this revision. It should now be possible to fully reproduce all major results presented in the revised manuscript.

Major Comment 8

Better articulate the specific research gaps being addressed. Although IP and privacy are mentioned as concerns, the manuscript lacks clear methods related to privacy protection, such as differential privacy or federated learning. Please clarify this connection or revise the discussion accordingly.

Author response to major comment 8: Thank you for your comment. This study is not directly concerned with user-level privacy modelling techniques such as differential privacy or federated learning, as it does not rely on sensitive, vehicle-internal, or user-identifiable data. The model uses only information routinely recorded by the charger during a session—specifically timestamped state of charge and power—without access to vehicle-side telemetry or behavioural identifiers. These data are provided via the standard EV–EVSE communication protocol, OCPP (Open Charge Point Protocol), and are generally agnostic to vehicle or user identity.

While certain metadata such as session or anonymous customer identifiers are technically present, the dataset used in this study contains no personally identifiable information. This design choice reduces exposure to privacy risks and supports broader applicability across infrastructure types. In a commercial deployment, customer IDs

(if accessible at session start by having a Shell account) could be leveraged to improve performance through cross-session information reuse or user-specific calibration, but such functionality is beyond the scope of the present work.

Regarding intellectual property (IP), it is important to note that the dataset as a whole, which includes SoC-power charging profiles, locations, and energy delivered across sessions, is considered commercially sensitive IP by Shell. Although individual sessions may not be sensitive in isolation, the complete dataset reveals operational patterns and infrastructure characteristics that are of strategic value.

While this study avoids vehicle-side telemetry and user identifiers, future extensions that incorporate cross-session linkage, personalised modelling, or behavioural inference would necessarily introduce new vectors for privacy risk. Addressing such challenges would require principled privacy preserving techniques such as differential privacy or federated learning not only to ensure compliance with data protection standards, but also to safeguard proprietary infrastructure usage patterns and commercial analytics pipelines.

We have added the following to the Discussion section of the revised manuscript:

Future work could include cross-session user modelling, personalised prediction, or behavioural inference. However, such directions would raise new privacy considerations, particularly if persistent identifiers or richer metadata are introduced. In these cases, privacy-preserving techniques such as differential privacy or federated learning may be necessary to ensure appropriate handling of both user-level data and commercially sensitive infrastructure information.

Major Comment 9

Clarify the logical relationship between the anomaly detection and the charging profile prediction modules to ensure coherence and methodological transparency.

Author response to major comment 9: Thank you very much for your comment, we acknowledge the significance of making clear the logical relationship between the anomaly detection and the charging profile prediction modules.

The underlying rationale is that, given the noisy and large-scale nature of the real-world dataset used in this work, it is essential to first remove clear outliers using the anomaly detection model. This step helps prevent the charging profile prediction model from being biased by anomalous or corrupted or unrepresentative sessions during training.

We have added additional explanations in the “anomaly detection and time series forecasting with β -variational autoencoder and temporal fusion transformer” subsection:

The deep learning workflow is designed for seamless integration with real-world deployment, with the model inputs consisting of anonymised information from charging stations. It contains two main components, a β -VAE-based anomaly detection model and a charging profile prediction model. Input charging profile data are structured as multivariate time series of charging power and SoC, augmented with static covariates including starting SoC, connector power rating, connector type, estimated EV battery capacity and ambient temperature. While the anomaly detection model requires complete

charging profiles to establish normative behaviour, the prediction model operates on partial profiles during real-time deployment, enabling adaptive forecasting. The static covariates selected are readily accessible: the charger provides information on starting SoC as well as connector type and power rating. Battery capacities of the EVs can be estimated dynamically by the charger as charging progresses or could be obtained through image recognition used to identify the EV model, while ambient temperature could be directly measured. However, in this work battery capacities were estimated using the total energy delivered during each session and the total change in SoCs, while ambient temperatures were obtained from ERA5 reanalysis data as described above. It is important to note that although the workflow is optimised for the inclusion of estimated EV battery capacity and ambient temperature, it remains capable of delivering reliable results without these features, with only a slight reduction in accuracy. Further details are provided in Supplementary Section S-3. An illustration of the proposed workflow, the anomaly detection model and the charging profile prediction model are shown in Figure 2.

Large scale real-world EV charging data are inherently noisy and inevitably contains spurious or corrupted measurement data, potentially rooting from a large number of sources, including connector misfits, firmware interruptions, charger malfunctions, data recording errors, or issues within the EV itself. While downstream predictive models must be robust to typical noise in real-world data, clearly abnormal sessions could lead to biases during both training and inference if not properly addressed. As these faults are not explicitly labelled in practice, a beta-variational autoencoder (β -VAE)-based anomaly detection model is trained on the complete set of full charging profiles in order to filter out spurious sessions and limit the influence of outliers. The β -VAE model is detailed in Figure 2b, and was trained to learn a compressed latent representation of normal charging profiles by minimising both the reconstruction loss and the Kullback-Leibler (KL) divergence loss. After training, each session is assigned a reconstruction-error score quantifying its conformity to the learned manifold, abnormal charging profiles are then detected as instances where the reconstruction loss exceeds a statistically derived threshold, signalling deviations from standard charging behaviour. Instances of detected abnormal charging sessions are presented in Figure 2a.

Major Comment 10

The approach of classifying anomalies based solely on reconstruction curve errors and charging time estimates may introduce substantial uncertainty due to the complexity of real-world conditions. Authors need to discuss and justify the robustness and validity of this anomaly classification.

Author response to major comment 10:

Thank you for your comment and we appreciate the opportunity to clarify. The anomaly detection model used in this study is not designed to classify anomalies in the conventional sense, nor does it produce labels for downstream evaluation. Rather, it functions as a preprocessing filter to identify sessions that are highly unlikely to be representative of regular charging behaviour, and excludes them from the training set. These sessions can behave erratically for a number of reasons, including connector misfits, firmware interruptions,

charger malfunctions, data recording errors, or issues within the EV itself, which are often unresolvable post hoc. This step is essential given the scale and heterogeneity of real-world datasets, where not all logged sessions are suitable for model training.

The criterion for exclusion is deliberately strict and based on large reconstruction errors and unrealistic inferred charging durations, in order to avoid removing valid data and thereby minimise false positives. These heuristics are used not to assign anomaly labels, but to prevent corrupted or non-standard sessions from degrading the quality of the predictive model. Since these sessions are excluded entirely from model training and evaluation, they do not introduce uncertainty into the learning process, but rather improve robustness by preventing spurious signals from being learned.

We have added additional explanations in the “Time to reliable prediction operationally post-plug-in” subsection:

Abnormal sessions were identified through reconstruction errors in the top percentile of the distribution and through cases where reconstructed charging durations exceeded actual values by more than 15 minutes. The temporal criterion specifically addresses pathological cases where power measurements transiently drop to zero, typically indicating sensor faults or disconnection events, which could artificially inflate predicted charging times and unfairly penalise model performance. The exclusion of these anomalous sessions ensures that training and evaluation reflects only physically plausible charging behaviour while preserving the overwhelming majority of statistically meaningful data. This minimal exclusion of 9,334 (1.02% of the full dataset) was carefully selected to remove only clear outliers without affecting the underlying data distribution. Following this anomaly exclusion step, the charging profile prediction model was trained on the filtered dataset and subsequently evaluated on two independent, previously unseen test sets.

Major Comment 11

Provide detailed analyses regarding potential information leakage or biases that could arise by excluding anomalous data during training.

Author response to major comment 11: Thank you for your comment and for the opportunity to clarify. The anomaly filter is applied prior to any model training or evaluation, and it is applied equally across all data partitions. Its purpose is not to select data based on target information or learned model outputs, but rather to remove sessions that exhibit clear signs of corruption or malfunction. These are sessions that would not be considered representative or reliable for training a data-driven model in any application. Importantly, excluding such sessions does *not* constitute information leakage, as no statistics from the test set are used in model training. Nor does this exclusion introduce bias, on the contrary, it reduces the risk of training the model on distorted data distributions. Including corrupted or unreliable data in training would pose a far greater threat to the validity of the model than excluding them transparently through this anomaly exclusion step. Moreover, it is worth noting that the number of sessions removed by this filter is very small (just over 1% of the dataset), so the impact on the overall data distribution is negligible. In practice, one would expect a more sophisticated and operationally integrated anomaly detection module to be used, tailored to specific use cases

and infrastructure. The current implementation serves primarily as a pragmatic screening step to ensure data quality in this proof-of-concept setting.

Major Comment 12

Enhance the evaluation rigor by performing ablation studies. Specifically, replace or remove specific model components to assess their individual contributions and overall model effectiveness.

Author response to major comment 12: Thank you for your comment. We have included both model architectural and feature ablation studies. The detailed results are represented in Figure 3, which corresponds to Figure 5 in the revised manuscript.

Figure 3: Model architecture and feature ablation studies. **c** Performance of the TFT-based charging profile prediction model compared against various baselines when evaluated on a combined dataset of test sets one and two. **d** Performance of the charging profile prediction model trained with different configurations of static covariates, evaluated with normalised MAE in prediction. The results integrate data from both test sets.

We have added the following text in the revised manuscript:

Figure 5c presents both a baseline comparison and an ablation study of model architecture. The TFT achieves the lowest normalised MAE among all models, outperforming recurrent neural networks (RNN), gated recurrent units (GRU), LSTM and the vanilla Transformer when evaluated on a combined dataset of test sets one and two. To isolate the contributions of key architectural components, two reduced variants are evaluated: removing the multi-head attention and feed forward linear modules yields the variable selection network plus LSTM (VSN-LSTM), while removing the recurrent components reduces the model to a pure Transformer. The superior performance of TFT over both simplified variants indicates that attention and recurrence provide complementary advantages for learning meaningful representations of charging profiles. Given the self-supervised nature of the task and the variability in input sequence lengths, comparison with traditional statistical learning approaches is not applicable, as such methods typically assume fixed-length feature vectors and

supervised targets. TFT was evaluated across three independent random seeds and data splits, despite the computational cost of 72 NVIDIA A100 GPU hours per training run. This robustness check was applied selectively to this model to validate the stability and generalisability of its performance, whereas the baselines were evaluated once under fixed conditions for efficiency.

The role of static covariates in the charging profile prediction model is examined through a series of feature ablation studies, with results shown in Figure 5d. These experiments assess the model’s performance when trained with all five static covariates compared to versions in which specific covariates are removed. The findings indicate that removing either estimated battery capacity or ambient temperature slightly degrades predictive accuracy, particularly when the model is given limited time series input. The effect is more pronounced for estimated capacity, which provides crucial contextual information in the early stages of a charging session. However, as more temporal data become available, the performance differences between the full and ablated models diminish considerably. This suggests that static covariates are especially valuable for early predictions, compensating for the absence of sufficient time-dependent structure, but their marginal contribution decreases as the charging profile becomes more fully observed. Additional methodological details regarding these ablation studies are provided in Supplementary Section S-3.

Major Comment 13

Incorporate comprehensive comparisons with baseline models such as LSTM variants, recent deep learning methods like other Transformer variants, and traditional time-series forecasting techniques. Such comparisons are essential to convincingly demonstrate the proposed model’s superiority and generalizability.

Author response to major comment 13: Thank you for your comment. As shown in Figure 3c in **Author response to major comment 12** (corresponding to Figure 5 in the revised manuscript), we include comparisons against several representative baselines, including recurrent models (RNN, GRU, LSTM), as well as a standard Transformer. These comparisons were conducted under identical training conditions, using the same input features and loss formulation, to ensure a fair evaluation.

We note that traditional time-series forecasting methods such as ARIMA, exponential smoothing, or classical regression-based techniques and statistical learning methods are not directly applicable in our setting. As the problem formulation in this work involves a self-supervised learning framework as well as making predictions over variable-length sequences using inputs that may consist of as little as a single point, it is structurally different from conventional time-series forecasting problems.

Importantly, we emphasise that this work is not only a model-centric study. Rather, it introduces a generalisable framework that enables real-world charging profile prediction using standard, charger-side data across a wide array of EV and infrastructure configurations, at scale. The aim is not to incrementally outperform baselines, but to demonstrate that accurate, real-world charging profile prediction is feasible under practical constraints. The value of the proposed framework lies as much in its generalisability, scalability, and deployment readiness as in its raw predictive accuracy.

Minor Comment 1

The model design is interesting, but lacks sufficient detail. It is recommended to provide detailed model parameters and training hyperparameters, supplement with pseudocode or a model flowchart, and elaborate on data preprocessing and feature engineering

Author response to minor comment 1: Thank you for your comment. The model architecture and workflow are illustrated in Figure 2 of the revised manuscript, and key model and training hyperparameters are detailed in the Methods section. For full transparency and reproducibility, the complete implementation has been submitted alongside this manuscript. The codebase includes all configuration files (in YAML format) that define the model architecture and training settings used to generate the results presented. It also contains the full data preprocessing pipeline and associated scripts. Feature engineering was intentionally kept minimal to promote model generalisability, and all relevant steps are clearly documented within the code.

Minor Comment 2

The manuscript mentions "1% reconstruction error and a 15-minute charging time difference as the anomaly detection threshold." Is there any theoretical basis for this? It is recommended to further discuss the rationality of the threshold.

Author response to minor comment 2: Thank you for your comment. The thresholds used for anomaly filtering, i.e. a reconstruction error in the top percentile and a reconstructed charging duration exceeding the actual value by more than 15 minutes, were selected as pragmatic heuristics. These criteria are not intended to serve as formal classification labels, but rather to identify and exclude sessions that could plausibly distort downstream model learning. The purpose here is not to detect anomalies as an end in itself, but to safeguard the charging profile prediction model from training on potentially corrupted or physically implausible sessions, such as those involving transient power dropouts or logging artefacts. In this sense, we are not seeking a theoretically optimal threshold, but a conservative one that prevents rare edge cases from disproportionately influencing model performance. We fully acknowledge that alternative strategies or more sophisticated anomaly detection approaches may be warranted in future work, especially for production-grade deployments. The thresholds we adopt are intentionally simple, interpretable, and easy to audit. Moreover, this filtering step affects only 1.02% of the dataset (9,334 out of close to 1 million sessions). We hope this clarifies that the anomaly criteria are not presented as definitive or theoretically grounded, but as practical measures to ensure training on valid, representative charging behaviour.

We have added additional explanations in the "Time to reliable prediction operationally post-plug-in" subsection:

Abnormal sessions were identified through reconstruction errors in the top percentile of the distribution and through cases where reconstructed charging durations exceeded actual values by more than 15 minutes. The temporal criterion specifically addresses pathological cases where power measurements transiently drop to zero, typically indicating sensor faults or disconnection events, which could artificially inflate predicted charging times and unfairly penalise model performance. The exclusion of these anomalous sessions ensures that training and evaluation reflects only physically plausible

charging behaviour while preserving the overwhelming majority of statistically meaningful data. This minimal exclusion of 9,334 (1.02% of the full dataset) was carefully selected to remove only clear outliers without affecting the underlying data distribution. Following this anomaly exclusion step, the charging profile prediction model was trained on the filtered dataset and subsequently evaluated on two independent, previously unseen test sets.

Minor Comment 3

The model’s inputs include charging power and battery SoC sequences. How is the SoC in this paper obtained, and how is the accuracy of the SoC ensured? Additionally, does the SoC have any impact on the deep learning method proposed in this paper?

Author response to minor comment 3: Thank you for your comment and the insightful questions. In this study, the state of charge (SoC) values are obtained directly from the charger logs via the Open Charge Point Protocol (OCPP) 1.6 standard, which defines SoC as a percentage reported by the vehicle to the charger. This value corresponds to the SoC displayed on the vehicle dashboard and reflects the vehicle’s internal estimate of remaining battery capacity. Although this is not necessarily the “true” SoC in a physical or electrochemical sense, e.g., some EVs retain a small energy reserve even when SoC reports 0%, much like fuel reserves in ICE vehicles, the estimation logic is implemented consistently for a given model by the OEM at production time.

Regarding accuracy and precision: the OCPP standard reports SoC as an integer percentage. For example, a reading of 5% is distinguishable from 4%, but values such as 4.4% are not transmitted or displayed. The SoC signal, therefore, has coarse precision but sufficient granularity for the temporal and behavioural patterns of interest in this study.

Repeatability of SoC values can vary due to underlying changes in effective battery capacity caused by temperature, age, and other unobservable factors. However, manufacturers do not expose these corrections, as they are treated as commercially sensitive IP. From the perspective of a charging network operator, the reported SoC is the only accessible signal, and its absolute accuracy or internal calibration is secondary to its consistent availability and predictive utility. In this sense, our model is designed to make the best possible use of the operational data available in real-world deployment settings.

From a machine learning perspective, SoC plays multiple roles. First, the SoC sequence serves to contextualise the charging power time series, which functions analogously to positional encoding in sequence models, helping to differentiate between charging behaviour at early vs. late stages of the session. Second, the starting SoC is included as a static covariate and has been shown (please see Figure 5 in the revised manuscript) to have a high learned feature importance score. This result is consistent with the fact that starting SoC strongly conditions the expected shape a charging profile, particularly in fast charging contexts where tapering effects dominate at high SoC levels.

Minor Comment 4

Since the dataset used in the manuscript contains proprietary information and is subject to confidentiality agreements, the authors do not disclose the data. Therefore, it is recommended to provide some non-confidential data or supplement the article with a more detailed description of the data to facilitate learning and verification by other researchers.

Author response to minor comment 4: Thank you for your comment. As stated in **Author response to major comment 7**, Shell UK has agreed to provide reviewers access to the dataset under strict confidentiality. The full anonymised dataset and pre-trained models are now included in this revision. It should now be possible to fully reproduce all major results presented in the revised manuscript. Additionally, a more detailed description of the dataset has now also been included in Figure 1 of the revised manuscript.

Minor Comment 5

The design of the figures in the manuscript is good, but the text in some figures is too small. It is recommended to ensure that the text is clear at 100% view, in accordance with the journal's requirements.

Author response to minor comment 5: Thank you for your comment and we appreciate your feedback regarding figure readability. At this stage, figures have been prepared according to the flexible formatting guidelines permitted during the review process. We acknowledge that the final production layout may differ substantially, and a separate formatting round is expected assuming acceptance. We commit to work closely with the editorial and production teams at that stage to ensure that all figure text, labels, and annotations meet the journal's final specifications and are clearly legible at 100% view, as required.

Minor Comment 6

The manuscript mentions that "The grouping for Nissan 315 Ariya/Leaf is less obvious." It is recommended to conduct an in-depth analysis of the reasons for similar situations.

Author response to minor comment 6: Thank you for your comment. In this revision, we use an updated model checkpoint with RevIN. In Figure 4 (Figure 5 in the revised manuscript), real-world charging sessions were screened and compared against a set of reference charging profiles. The reference curves, generated through experiments conducted by Shell under controlled laboratory conditions, encompass a diverse range of EV brands, models, connector power ratings, connector types, and ambient temperatures. A curve-matching algorithm implemented by Shell was used to identify the closest matches between the real-world profiles and the reference curves. Around 15 thousand charging profiles with more than 60 data points and exhibiting a strong match with a reference curve were subsequently compressed into latent representations by the β -VAE encoder, projected onto a 2D space using t-distributed stochastic neighbour embedding (t-SNE). It is important to note that while the curve-matching process is robust, it is a perfect reflection of reality, and as some of the reference charging

profiles are highly similar (for instance, Nissan Leaf at 30 degrees Celsius with 50 kW reference charging profile very closely resembles BMW i3 also at 30 degrees Celsius and 50 kW), they can lead to visually overlapping clusters. The limited separation between some clusters arises from two factors: (1) the β -VAE exhibits a degree of scale invariance due to the RevIN module, which causes it to prioritise curve shape over absolute magnitude; and (2) several reference profiles are themselves highly similar in form, making it difficult to distinguish between them even under ideal matching. In this particular case, the grouping of Nissan Ariya Leaf is less obvious because its reference profile closely resembles that of other brands, including BMW i3.

Figure 4: Model interpretations highlighting the knowledge learnt by the machine learning models, as well as ablation studies on the charging profile prediction model. **a** t-SNE visualisation of the latent representations of charging data. Each point represents a full charging profile, coloured according to its closest match with a reference profile obtained through charging under ideal laboratory conditions. **b** Zoomed-in views of the clusters formed in the t-SNE plot, showing detailed charging profiles selected from these clusters alongside their corresponding matched reference profiles.

Minor Comment 7

Please discuss economic feasibility of the method and uncertainty analysis, as they are crucial for practical deployment conditions.

Author response to minor comment 7: Thank you for your comment and for highlighting the importance of practical deployment considerations. We agree that economic feasibility and uncertainty quantification are important topics, especially in large-scale operational contexts. However, we would like to clarify that the focus of this work is on establishing a technically robust and generalisable predictive framework for real-world EV fast charging profiles, using standard charger-side data. As such, a detailed economic analysis (e.g., modelling costs across cloud infrastructure, deployment pipelines, or edge devices) lies outside the present scope and would likely require significant variation across use cases and infrastructure ownership models.

That said, the inference benchmarks we provide (please see Table 3 in **Author response to major comment 6**)

show that forward passes are extremely fast (on the order of 0.1 seconds for over 1,000 sessions on standard GPU hardware). This indicates that, from a computational standpoint, real-time deployment is readily achievable at negligible marginal cost. The model could be hosted centrally and accessed via lightweight API calls, or potentially embedded in local systems depending on specific use cases and resource constraints.

From a commercial perspective, efficient scheduling of vehicles is a long-standing concern for fleet operators, with established markets for route-optimisation services and logistics platforms. Similarly, Shell has a clear operational interest in efficient scheduling and throughput optimisation across its charge post network. This work directly contributes to that broader effort by enabling fast, data-driven prediction of session durations.

In terms of uncertainty, the model already incorporates a degree of predictive uncertainty by training with a quantile loss objective. This allows it to estimate conditional quantiles of the target distribution (e.g., 10th, 25th, 50th, 75th, and 90th percentiles), rather than just a single point prediction. These quantile predictions provide a practical and interpretable way to express uncertainty in forecasted charging durations or profiles. While not a fully probabilistic framework, this approach enables meaningful uncertainty-aware predictions.

Minor Comment 8

The anomaly detection process filters data based on reconstruction error and charging time discrepancies (>15 minutes), but the rationale behind the threshold selection (e.g., whether it is based on a quantile of the data distribution) is not provided. Clarifying this aspect would improve the model’s transparency and reliability.

Author response to minor comment 8: Thank you for your comment and the opportunity to clarify. The reconstruction-error-based anomaly exclusion is indeed quantile-based: we excluded sessions whose reconstruction error fell in the top 1% of the distribution. This threshold was selected to remove statistical outliers while preserving the overwhelming majority of the dataset.

The secondary exclusion criterion—removing sessions where the reconstructed charging duration exceeded the actual by more than 15 minutes—was not derived from a formal statistical basis but was instead chosen based on domain knowledge and empirical inspection of the data. These sessions (as shown in Figure 3a in the revised manuscript) typically reflect pathological cases such as transient sensor faults, dropped power readings, which would otherwise introduce distortions in training or evaluation. This rule affected only a small number of sessions (a few hundred), and its impact on the overall data distribution is negligible.

We would also like to clarify that this step is not a general-purpose anomaly detection algorithm intended for evaluation of labelled anomalies. Rather, it is a pragmatic data cleaning/preprocessing step designed to prevent corrupted or physically implausible sessions from impairing model training. More sophisticated anomaly detection schemes could certainly be developed for this setting, but they lie outside the scope of the current study.

We have added additional explanations in the “Time to reliable prediction operationally post-plug-in” subsection:

Abnormal sessions were identified through reconstruction errors in the top percentile of the distribution and through cases where reconstructed charging durations exceeded actual values by more than 15

minutes. The temporal criterion specifically addresses pathological cases where power measurements transiently drop to zero, typically indicating sensor faults or disconnection events, which could artificially inflate predicted charging times and unfairly penalise model performance. The exclusion of these anomalous sessions ensures that training and evaluation reflects only physically plausible charging behaviour while preserving the overwhelming majority of statistically meaningful data. This minimal exclusion of 9,334 (1.02% of the full dataset) was carefully selected to remove only clear outliers without affecting the underlying data distribution. Following this anomaly exclusion step, the charging profile prediction model was trained on the filtered dataset and subsequently evaluated on two independent, previously unseen test sets.

Minor Comment 9

The dataset is limited to Europe’s temperate climate (-14°C to 35°C). A discussion on the model’s applicability in extreme climates (e.g., tropical or polar regions) or experiments involving cross-region transfer learning would enhance the work’s global relevance.

Author response to minor comment 9: Thank you for your thoughtful comment. We agree that evaluating model performance under extreme climate conditions, such as tropical or polar environments, would strengthen the global relevance of this work. However, the current study is necessarily limited by the availability of data, which includes over 900,000 real-world DC fast charging sessions recorded across the UK and Germany, covering ambient temperatures between -14°C and 35°C . While this range reflects typical usage across much of Western Europe, it does not extend to more extreme climatic regions. It is worth noting, however, that the vast majority of EVs currently in operation are located in regions where such temperature ranges prevail for most of the year. As a result, the scope of the present study remains commercially and operationally relevant to the dominant share of global EV activity.

Nonetheless, we have taken steps to evaluate the model’s geographic robustness within the scope of the available dataset. In particular, we performed a zero-shot evaluation on a held-out test set comprising charging sessions from the Netherlands—data that were not used during model training or hyperparameter tuning. This subset also lacked the estimated battery capacity input, and we therefore used a model variant trained without this feature. As shown in Figure 5f of the revised manuscript, the model achieved comparable accuracy on the Netherlands data relative to both test set 1 (random split) and test set 2 (temporal split), demonstrating generalisation to geographical regions within Western Europe.

We recognise that transfer learning and domain adaptation to more extreme climates would be a valuable extension, and we consider this an important direction for future work—particularly as more diverse global datasets become available.

Figure 5: Model evaluation on held-out test sessions from the Netherlands.

We have added additional explanations in the “Model-derived insights into real-world charging behaviours” subsection:

To assess the model’s ability to generalise geographically, an evaluation was conducted on an additional hold-out test set comprising charging sessions only from the Netherlands, which was not included in training. As this dataset lacked battery capacity information, the model trained without estimated capacity was used. As shown in Figure 5f, this model achieved comparable performance on the Netherlands data and the primary test sets from the UK and Germany, indicating a degree of robustness to geographic distribution shift within Western Europe, even under missing covariates.

Minor Comment 10

While the paper includes a variable importance analysis, a deeper exploration of the physical interpretability of key factors (e.g., battery capacity) would strengthen the analysis and improve the model’s transparency.

Author response to minor comment 10: Thank you for your insightful comment. We also find the physical interpretability of static covariates to be an interesting topic. We think that as with the current setup of static covariates, estimated battery capacity is the only feature that can be used to differentiate among different EV models, at could be used by the model implicitly as a surrogate for EV types or battery chemistry. We did additional experiments reducing the granularity of the continuous estimated battery capacity covariate and trained the model with a discretised version of the variable. The results are shown in Figure 8 in **Author response to minor comment 9** (or Figure 5f in the revised manuscript). We showed that reducing the granularity of the continuous estimated battery capacity covariate led to reduced performance, which might signal that this variable may encode more information than nominal pack size.

We have added the following to the “Time to reliable prediction operationally post-plug-in” subsection:

Figure 5f also presents an ablation study examining the effect of reducing the granularity of the estimated capacity input. Rather than using continuous values, estimated battery capacity was discretised into four broad categories (10 to 50, 50 to 80, 80 to 120, and more than 120 kWh). This binned representation yielded intermediate performance, outperforming the model trained without capacity but underperforming relative to the continuous variant. These performance differences suggest that estimated capacity conveys more than nominal pack size, it may implicitly encode additional battery-specific characteristics, such as cell chemistry or charging rate limitations. Given that it is the only static input varying systematically across EVs, while other inputs primarily reflect session-specific conditions, this covariate likely acts as a proxy for EV identity, enabling the model to infer latent structural or electrochemical differences. While such associations cannot be directly verified from the available data, the observed performance degradation upon discretisation supports the view that preserving the continuous representation allows the model to exploit fine-grained distinctions relevant to real-world charging behaviour.

Authors' Response to Reviewer 2

General Comments. I co-reviewed this manuscript with one of the reviewers who provided the listed reports. This is part of the Nature Communications initiative to facilitate training in peer review and to provide appropriate recognition for Early Career Researchers who co-review manuscripts.

Response: Thank you for your time and involvement in the review process. We appreciate your contribution through the co-review and hope the revised manuscript meets expectations.

Comment 1

The reviewer can see the code but without the data. Data needs to be seen very clearly.

Response: Thank you for your comment. Shell UK has agreed to provide reviewers access to the dataset under strict confidentiality. The full anonymised dataset and pre-trained models are now included in this revision. It should now be possible to fully reproduce all major results presented in this manuscript.

Authors' Response to Reviewer 3

General Comments. This paper presents a deep learning model that predicts EV charging profiles and durations using minimal input from over 900,000 real-world DC fast charging sessions. The model operates in real time, provides uncertainty estimates, and achieves up to 95% accuracy with data collected within the first five minutes of charging. The topic is timely and relevant for practical EV charging applications. Notably, the model demonstrates the generalization across different EV models and delivers high prediction accuracy early in the charging process. Overall, this paper is not only well-organized and is highly valuable for the EV applications.

Author response to general comments: Thank you for your thoughtful and constructive review. We are especially grateful for your recognition of the paper's contribution and its relevance to real-world EV charging applications. Your insightful questions have helped us further clarify and strengthen key aspects of the manuscript and we are grateful for the time and effort you dedicated to this review. We believe we have addressed all the comments item by item as follows.

Comment 1

In Figure 2a, the proposed framework is shown to dynamically adapt to real-time data using a cloud server and HPC platform. Please provide comments on the computational load and feasibility for practical, large-scale deployment.

Author response to comment 1: Thank you for your comment and for raising this crucial point for improving the practical focus of this manuscript. We have now included detailed inference benchmarking results in the revised manuscript (Table 3), evaluating the computational load under varying batch sizes on both CPU and GPU. These results demonstrate that a single forward pass over 1,024 sessions completes in approximately 100 milliseconds on a single GPU, corresponding to a throughput exceeding 10,000 sessions per second. Even with CPU-only execution, inference remains tractable at smaller batch sizes, making local or edge deployment plausible for low-volume scenarios.

In terms of practical feasibility, this high throughput far exceeds the temporal resolution of charger-side input data (typically one update every 30–60 seconds at most), indicating that the model is well-suited to real-time operation in large-scale deployments. Moreover, the compact model architecture and event-driven inference design enable efficient resource usage, allowing deployment via standard cloud infrastructure or lightweight microservices architecture, without requiring persistent GPU access.

Although training requires approximately 72 A100 GPU hours per run, this cost is incurred infrequently. In practical deployment, retraining would occur periodically (e.g., weekly or monthly) to incorporate new data and adapt to operational drift, and could be scheduled using standard cloud compute resources without disrupting real-time service.

We have added the following to Supplementary information in the main text:

Inference benchmarks on the charging profile prediction model are conducted on a dedicated compute node configured with an AMD EPYC 7742 64-core processor and a single 40 GB NVIDIA A100

GPU. The benchmarking evaluates the inference efficiency of the model under varying batch sizes, comparing CPU-only and GPU-accelerated execution timing, with the timing statistics reported over 50 runs. The results are displayed in Table 3.

Table 2: Forward pass timing and throughput across batch sizes on CPU and GPU.

Batch Size	CPU Time (ms)	GPU Time (ms)	CPU Throughput (samples/sec)	GPU Throughput (samples/sec)
1	37.17 ± 0.46	12.13 ± 0.04	26.9	82.4
16	309.91 ± 3.83	12.72 ± 0.02	51.6	1257.5
32	625.27 ± 28.35	13.24 ± 0.06	51.2	2416.2
64	1302.49 ± 42.44	15.39 ± 0.04	49.1	4158.0
128	2759.00 ± 55.99	20.36 ± 1.58	46.4	6287.4
256	5596.13 ± 82.89	32.63 ± 0.11	45.7	7844.4
512	12126.96 ± 130.86	54.69 ± 0.20	42.2	9361.8
1024	24577.94 ± 117.24	101.08 ± 0.26	41.7	10130.7

We have added the following to the discussion section:

The model is designed for practical deployment. It operates in an event-driven fashion, updating predictions as new SoC readings are received, typically every one minute. Inference is computationally efficient: a single forward pass over 1,024 sessions completes in approximately 100 milliseconds on standard GPU hardware (Table 3), corresponding to a throughput of over 10,000 sessions per second. This is several orders of magnitude faster than the rate at which new input data becomes available, confirming that the method is capable of real-time operation even at scale.

Comment 2

Would battery chemistry be considered an important covariate in the model? A discussion on this aspect would be helpful, particularly considering the variability across different EV models.

Author response to comment 2: Thank you very much for your insightful comment. Indeed, we think battery chemistry is a highly important covariate in the model. Although battery chemistry information is not currently communicated through the charging network under existing standards and hence not included as an explicit input to the model, we believe the model may be capturing its influence implicitly through static covariates such as estimated battery capacity, as well as through patterns in the charging time series. That being said, the ongoing development of battery passport legislation within the European Union may make such information accessible in the future. Should this become available, it could be readily incorporated as an additional covariate in the framework and we expect this to offer meaningful performance gains.

Comment 3

Please elaborate on how the influence of covariates might change as the dataset becomes more diverse, especially in terms of EV types, battery chemistries, and user charging behaviors.

Author response to comment 3: Thank you for raising this thought-provoking question, it prompted us to consider the covariate importance more deeply. As the dataset becomes even more diverse, incorporating a wider range of EV types, battery chemistries, charging hardware, and user behaviours, we expect the relative influence of existing covariates to continue to evolve.

We expect the estimated battery capacity feature to become even more important. The first reason being that with the current setup of static covariates, estimated battery capacity is the only feature that can be used to differentiate among different EV models. We believe that it may already be serving as a surrogate for different EV models and battery chemistries. As more EV types are represented in the dataset, especially with overlapping charging behaviours or power levels, the model will rely more heavily on features that help distinguish them. Estimated battery capacity will naturally gain prominence in such settings because it anchors the model's understanding of what "normal" charging looks like for each underlying vehicle configuration. This feature might also be able to interact more meaningfully with other covariates, such as the starting SoC. For instance, 50% starting SoC on a 90 kWh battery might imply a very different charging speed compared to 50% on a 40 kWh battery. The model might be able to exploit this relationship more effectively with increased vehicle diversity.

We expect ambient temperature to become more important. We think that the current feature importance score does not give enough credit to the ambient temperature, maybe due to its high correlation with the charging power at the beginning of sessions. But with data from more extreme climates or seasonal conditions being included, the model should be able to learn to pay more attention to this feature. That said, improvements in EV thermal management may moderate this trend: early EVs often lacked sophisticated temperature regulation systems, whereas modern designs increasingly integrate more effective and affordable cooling and heating strategies. As battery thermal management continues to improve across newer vehicle models, the direct influence of ambient temperature on charging behaviour may become less pronounced in practice.

Charger type has so far exhibited relatively low importance. This is not unexpected, as connector type is often highly correlated with charger power. Nonetheless, as the infrastructure landscape diversifies, for example, with increasing deployment of ultra-fast chargers or mixed-standard stations—this covariate may carry more weight in future iterations of the model.

The model architecture and learning framework are designed to adapt to these dynamics automatically. Because feature importance is learned end-to-end during training, the system remains robust and scalable as the training distribution evolves, without the need for manual feature re-engineering.

Authors' Response to Reviewer 4

General Comments. The topic of this paper is of high relevance. The performance of the novel method is thoroughly demonstrated and proved here. The overall structure of the paper is coherent. In the following, I will elaborate on different aspects that would improve the scientific quality of this manuscript and also the understanding to the reader..

Author response to general comments: Thank you very much for your thoughtful and constructive review. We sincerely appreciate your recognition of the relevance of the topic, the coherence of the manuscript, and the demonstrated performance of the proposed method. Your detailed comments and suggestions were both insightful and highly constructive, and they have played an important role in improving the scientific quality and clarity of the revised manuscript. We are grateful for the time and effort you dedicated to this review. We believe we have addressed all the comments item by item as follows.

Comment 1

Title: The title is very strong. I suggest to include also "duration", f.e. "AI predicts real-world EV DC charging profiles and charging duration with minimal inputs"

Author response to comment 1: Thank you very much for your suggestion. We have revised the title as recommended. The new title is: "AI predicts real-world EV DC charging profiles and durations with minimal inputs."

Comment 2

In Main: - The state of the art discussion is missing. The reader should get some idea of what the current methods in this field are capable of in terms of the performance parameters that are novel in your approach.

Author response to comment 2: Thank you for your comment. As far as we are aware, this work is the first to develop and evaluate a charging profile prediction model using a large-scale, real-world dataset comprising over 900,000 public DC fast charging sessions. Prior approaches have typically relied on small-scale laboratory experiments, simulations, or proprietary vehicle telemetry—often involving tightly controlled conditions and limited diversity in vehicle and charger types.

The most relevant existing works include Tian et al. [7], which focuses on laboratory charging profiles of individual lithium-ion cells, and Shi et al. [1], which investigates charging duration prediction using BMS data under relatively idealised conditions without accounting for variability across EVs or charger configurations. In contrast, our framework operates in a real-world deployment setting, using only standard charger-side data, and accommodates substantial heterogeneity in vehicles, infrastructure, and usage patterns. Our proposed

framework also introduces novel capabilities in this context, including real-time updates, uncertainty estimation, and minimal input requirements, none of which are standard features in existing approaches.

We have added the following to the Introduction section:

Recent advances in data-driven modelling of lithium-ion battery systems have increasingly addressed challenges associated with random, scarce, and heterogeneous data using sophisticated machine learning techniques [2, 5, 4, 3]. These studies, often based on real-world datasets, have achieved strong performance in tasks such as state of health estimation, lifetime prediction, and retired battery sorting, and represent meaningful progress toward the development of robust and generalisable battery analytics. By contrast, research specifically focused on predicting EV charging profiles remains comparatively limited. Existing work in this area has largely relied on relatively small datasets, often collected under laboratory or semi-controlled conditions using instrumented cells or proprietary battery management system (BMS) data [6, 7, 1, 8, 9]. Such data are frequently tied to specific vehicles and may not be publicly accessible due to design constraints and privacy concerns specific to original equipment manufacturers (OEMs), which limits the broader applicability and scalability of these approaches. Moreover, while the use of real-world EV data is becoming more common, it has predominantly been applied to tasks such as state-of-health estimation, fault detection, or lifetime prediction [10, 11, 12, 13, 14], rather than to the forward-looking task of charging profile prediction, despite its relevance for smart charging, range forecasting, charger utilisation planning, and energy services.

Comment 3

In Main: - Why use predictive models and not a physical one? What are the downsides of using the relevant parameters and compute a physical model?

Author response to comment 3: Thank you for your insightful comment. This work is intentionally built on charger-side data only, without access to internal vehicle telemetry or battery management system (BMS) parameters. While physics-based models can be constructed using BMS data, potentially offering higher fidelity in idealised conditions, such models typically require detailed knowledge of battery chemistries, internal states, and thermal dynamics, which are not available in public DC fast charging settings. Moreover, physical models often involve complex system identification and parameter tuning per vehicle type, making them difficult to scale or generalise across the diversity of real-world EVs and chargers.

In contrast, this study demonstrates that data-driven models can achieve strong predictive performance using only standardised inputs available at the charger. The results show that even without access to internal battery states, accurate charging duration and profile predictions are feasible at scale.

In future works, hybrid approaches that combine the merits of both physical modelling and data-driven methods such as digital twins or physics-informed machine learning could offer further improvements, combining the strengths of both paradigms where additional information is available.

Comment 4

In Main: - One of the interesting aspects is the anomaly detection; the authors should introduce this concept in more proper way in the beginning of the text and than later also go into typical applications for forecasting of this and why this is essential to the approach.

Author response to comment 4: Thank you very much for your comment, in this revision we have added a more complete introduction to the anomaly detection model and made clear its relationship with the charging profile prediction model. We have clarified that the primary intention for having this anomaly detection model is to try to limit the influence of spurious charging sessions on the training of the charging profile prediction model. The underlying rationale is that, given the noisy and large-scale nature of the real-world dataset used in this work, it is essential to first remove clear outliers using the anomaly detection model. This step helps prevent the charging profile prediction model from being biased by anomalous or unrepresentative sessions during training. We would also like to clarify that this step is not a general-purpose anomaly detection algorithm intended for evaluation of labelled anomalies. Rather, it is a pragmatic data cleaning/preprocessing step designed to prevent corrupted or physically implausible sessions from impairing model training. More sophisticated anomaly detection schemes could certainly be developed for this setting, but they lie outside the scope of the current study.

We have added additional explanations in the “anomaly detection and time series forecasting with β -variational autoencoder and temporal fusion transformer” subsection:

The deep learning workflow is designed for seamless integration with real-world deployment, with the model inputs consisting of anonymised information from charging stations. It contains two main components, a β -VAE-based anomaly detection model and a charging profile prediction model. Input charging profile data are structured as multivariate time series of charging power and SoC, augmented with static covariates including starting SoC, connector power rating, connector type, estimated EV battery capacity and ambient temperature. While the anomaly detection model requires complete charging profiles to establish normative behaviour, the prediction model operates on partial profiles during real-time deployment, enabling adaptive forecasting. The static covariates selected are readily accessible: the charger provides information on starting SoC as well as connector type and power rating. Battery capacities of the EVs can be estimated dynamically by the charger as charging progresses or could be obtained through image recognition used to identify the EV model, while ambient temperature could be directly measured. However, in this work battery capacities were estimated using the total energy delivered during each session and the total change in SoCs, while ambient temperatures were obtained from ERA5 reanalysis data as described above. It is important to note that although the workflow is optimised for the inclusion of estimated EV battery capacity and ambient temperature, it remains capable of delivering reliable results without these features, with only a slight reduction in accuracy. Further details are provided in Supplementary Section S-3. An illustration of the proposed workflow, the anomaly detection model and the charging profile prediction model are shown in Figure 2.

Large scale real-world EV charging data are inherently noisy and inevitably contains spurious or corrupted measurement data, potentially rooting from a large number of sources, including connector

misfits, firmware interruptions, charger malfunctions, data recording errors, or issues within the EV itself. While downstream predictive models must be robust to typical noise in real-world data, clearly abnormal sessions could lead to biases during both training and inference if not properly addressed. As these faults are not explicitly labelled in practice, a beta-variational autoencoder (β -VAE)-based anomaly detection model is trained on the complete set of full charging profiles in order to filter out spurious sessions and limit the influence of outliers. The β -VAE model is detailed in Figure 2b, and was trained to learn a compressed latent representation of normal charging profiles by minimising both the reconstruction loss and the Kullback-Leibler (KL) divergence loss. After training, each session is assigned a reconstruction-error score quantifying its conformity to the learned manifold, abnormal charging profiles are then detected as instances where the reconstruction loss exceeds a statistically derived threshold, signalling deviations from standard charging behaviour. Instances of detected abnormal charging sessions are presented in Figure 2a.

Comment 5

comments on the method description: Mention the temporal resolution. -This is more of a general comment. But this information should by already included in the abstract, then main, should be also part of the state of the art. There is no mention of the temporal resolution. But this is specifically when it comes to electricity market applications one of the core parameters, specifically for congestion managment or grid stability. The same goes for the prediction of the charging profile? How long does it take to generate this?

Author response to comment 5: Thank you for raising this important and highly relevant point. We agree that temporal resolution and inference time are critical aspects, especially in the context of electricity market applications such as congestion management and grid stability. This was indeed an oversight in the original draft, and we have now addressed it explicitly in the revised manuscript.

Regarding temporal resolution, we have now clarified that the model operates in an event-driven manner, updating its predictions whenever a new SoC reading becomes available, i.e. a single forward pass of the model is required when there is at least one percentage point change in SoC (because the SoC are only reported to the charger at integer values). The exact timing varies between sessions depending on factors such as starting SoC and charging power, we report the time requires to accumulate a specific number of SoC points in Figure 6 (Figure 4e in the revised manuscript). In practice, SoC and power data are reported periodically at fixed intervals (typically every 60 seconds), irrespective of whether the values themselves have changed. Companies like Shell commonly choose a 60-second interval as a practical compromise between data volume and operational efficiency. However, as the value of predictive tools such as the one presented here becomes better understood, these sampling intervals could be reconfigured dynamically. For example, high-traffic or operationally critical sites could report more frequently (e.g., every 30 seconds), enabling tighter prediction loops and greater responsiveness in time-sensitive contexts. As for inference time, we note that a forward pass over 1,024 sessions takes approximately 100 milliseconds on standard GPU hardware (see Table 3), confirming that real-time or near-real-time operation is computationally feasible at scale.

6

Figure 6: The median time required to accumulate a specific number of points on the charging profile, with error bars representing the IQR.

We have added the following paragraph to the results section under "Time to reliable prediction operationally post-plug-in" subsection:

It should be noted that while the charging profile data are logged by chargers at regular intervals of one minute, depending on the provider, the machine learning model is designed to update its predictions in an event-driven manner, specifically when the reported SoC increases by at least 1%, reflecting the integer precision defined by the Open Charge Point Protocol (OCPP) standard. Many operators commonly use a 60 second logging interval to balance data volume and utility, but this setting is configurable. For instance, busier or critical sites could report more frequently to enable more accurate predictions. Since DC fast charging is highly non-linear, the timing of these SoC updates is irregular and varies across sessions. Figure 1 shows the median elapsed time required to accumulate a given number of SoC points. Early in the charging process, multiple SoC updates are typically observed within a few minutes, while in later stages, longer durations are required due to reduced charging power. This results in a variable effective temporal resolution, governed by the charging dynamics rather than fixed time intervals. In practice, model updates may occur every minute in the early stages of a session, enabling the model to make timely and increasingly accurate predictions as more data becomes available. As the charging rate slows and updates become less frequent, the model’s predictions have typically already converged, reducing the urgency for further refinement.

Regarding inference time, we conducted experiments on a standard compute node equipped with both CPU and GPU to measure the time required for the charging profile prediction model to perform inference. We found that with standard GPU hardware, the charging profile prediction model can generate predictions for 10,000 sessions per second. We believe that this supports the claim that the model is suitable for real-time deployment, even at scale.

We have added the following to Supplementary information in the main text:

Inference benchmarks on the charging profile prediction model are conducted on a dedicated compute node configured with an AMD EPYC 7742 64-core processor and a single 40 GB NVIDIA A100 GPU. The benchmarking evaluates the inference efficiency of the model under varying batch sizes,

comparing CPU-only and GPU-accelerated execution timing, with the timing statistics reported over 50 runs. The results are displayed in Table 3.

Table 3: Forward pass timing and throughput across batch sizes on CPU and GPU.

Batch Size	CPU Time (ms)	GPU Time (ms)	CPU Throughput (samples/sec)	GPU Throughput (samples/sec)
1	37.17 ± 0.46	12.13 ± 0.04	26.9	82.4
16	309.91 ± 3.83	12.72 ± 0.02	51.6	1257.5
32	625.27 ± 28.35	13.24 ± 0.06	51.2	2416.2
64	1302.49 ± 42.44	15.39 ± 0.04	49.1	4158.0
128	2759.00 ± 55.99	20.36 ± 1.58	46.4	6287.4
256	5596.13 ± 82.89	32.63 ± 0.11	45.7	7844.4
512	12126.96 ± 130.86	54.69 ± 0.20	42.2	9361.8
1024	24577.94 ± 117.24	101.08 ± 0.26	41.7	10130.7

We have added the following to the Discussion section:

The model is designed for practical deployment. It operates in an event-driven fashion, updating predictions as new SoC readings are received, typically every one minute. Inference is computationally efficient: a single forward pass over 1,024 sessions completes in approximately 100 milliseconds on standard GPU hardware (Table 3), corresponding to a throughput of over 10,000 sessions per second. This is several orders of magnitude faster than the rate at which new input data becomes available, confirming that the method is capable of real-time operation even at scale.

We have rewritten the abstract to reflect this information:

Accurate prediction of electric vehicle (EV) charging profiles and durations is critical for improving EV adoption and optimising charging infrastructure. Direct current fast charging (DCFC) presents complex, diverse charging behaviours shaped by many interacting factors. This work introduces a deep learning framework trained on 909,135 real-world DCFC sessions, capable of predicting full charging profiles and durations from minimal input while providing uncertainty estimates. The model initiates predictions from a single point on the power/state-of-charge (SoC) profile and incrementally refines them as new SoC observations arrive, operating in an event-driven manner that enables real-time updates and efficient inference. Evaluated on previously unseen data under realistic conditions, the model generalises across vehicle types and charging scenarios. It achieves 90% average relative accuracy in predicting charging duration from a single point, and 95% accuracy with an absolute error under one minute using six points collected within five minutes of charging time.

Comment 6

Results: - you argue in 95% of cases, that there is a 1 minute error -> what exactly is meant of this? that the difference between the forecasted power level and between the actual is one minute? Or is this the duration? I would more expect that when charging profiles are predicted, the power level is also of interest. For energy-related applications, the error in power level is definitely more of interest.

Author response to comment 6: Thank you for your comment and we appreciate the opportunity to clarify. Since the charging profile prediction model produces outputs based on a variable number of input points, and can progressively refine its predictions as more data become available, we report accuracy metrics for both predicted power profiles and charging duration as a function of the number of input points used. These results are presented in Figure 7 (Figure 4a in the revised manuscript), covering input sizes from a single point up to 15. To make the results more interpretable, we have updated the metrics for predicting the charging profile as the normalised mean absolute error, which is the error in predicting charging power normalised by the nominal rated charger power. For instance, the leftmost panel of Figure 7 shows that on test set one, the mean error in predicted power when a single point on the charging profile is used as input is approximately 0.041 or 4.1%. The centre and right panels report the relative and absolute errors in predicting total charging duration, respectively. We observed that when six points are used as input to the charging profile prediction model, the accuracy metrics are: approximately 0.02 mean normalised MAE in predicting power, less than 5% mean relative error in predicting charging duration, and less than one minute mean absolute error in predicting charging duration. Hence we claimed in the abstract that “The model achieves 90% average relative accuracy in predicting charging duration from a single point, and 95% accuracy with an absolute error under one minute using six points collected within five minutes of charging time.” In this case, when six points are used as input, the error in predicting power is about 2%.

Figure 7: Performance evaluation of the charging profile prediction model, where the model’s median predictions were compared to ground truth charging profiles and times. Charging times were calculated based on either the completion of the charging session or reaching 80% SoC, whichever came first. Analysis was confined to sessions with a strictly greater than 15% change in SoC. **a** The charging profile prediction model’s performance on predicting the power-SoC charging profile and charging time for different numbers of known points of the charging profile as input, evaluated on the two test sets. Normalised mean absolute error (MAE) was computed as the MAE of predicted power normalised by connector power rating. The error bars represent the interquartile range (IQR).

Comment 7

Comments on the figures:

Figure 1d: I suggest to add 'estimated *battery* capacity' for better understanding

Figure 2: - nice workflow description - Maybe highlight the step of charging profile/time predictive which is the novelty that you contribute - also it seems that only a-c are really relevant /d-f do seem more detail - I suggest to make the plots b) and c) more readable by f.e. make the text horizontal instead of vertical

figure 3: (b) the three different cases are not understandable from the beginning; i suggest to add a description: example 1/2/3

Author response to comment 7: Thank you for your comment and for your detailed suggestions.

Figure 1d: We have added estimated battery capacity as you suggested.

Figure 2: You are entirely correct that d–f are more detailed architecture. We feel that these are some of the more lesser known architectures to use and in order to help the readers we included their schematics for completeness. We agree that b and c should be made more readable, we prepared another version of this that makes the text horizontal, however that ended up being too wide for this particular formatting. Depending on the final layout in the formatting round (which will most likely be double-column), we will decide then which one is best to use.

Figure 3: We have added a description as you suggested to the Figure captions.

Comment 8

Discussion: I expected stronger points to be made in the discussion. Currently, the discussion rather is a summary. The following points should be addressed: - What applications can this new minimal-input method with the given performance parameters enhance? The relevancy for dso and tso operation; as well as consumer-side benefits and applications? - What does this method allow to improve in these applications? (compared to the current state of the art)

Author response to comment 8: Thank you for your constructive comment. We agree that a more pointed discussion of the model's applications and its implications relative to current methods strengthens the manuscript. We have revised the Discussion section accordingly to directly address the roles this minimal-input framework can play in practical applications, including distribution and transmission system operator (DSO/TSO) use cases and consumer-facing services.

Regarding comparison with the current state of the art: In the publicly available academic literature, to the best of our knowledge, this work represents the first large-scale, real-world demonstration of charging profile prediction using over 900,000 heterogeneous public DCFC sessions. Previous studies that focus on charging profile prediction typically rely on small laboratory datasets, simulated charging sessions, or vehicle-side telemetry data under idealised conditions, often with limited scale and generalisability. In the commercial sector, the most advanced solutions are not openly published and remain proprietary. While it is difficult to formally benchmark against these systems, we are not aware of any publicly demonstrated commercial approach that offers comparable predictive accuracy, input minimalism, and scalability across EV types and charging infrastructures.

In terms of specific DSO and TSO applications, this method enhances the ability to forecast site-level demand based on real-time session observations. This is particularly valuable for local congestion management, short-term load forecasting, and dynamic grid services. TSOs and DSOs are not only concerned with the aggregate load across all active charge points at a site, but also with the interaction between charging demand, on-site energy buffering systems (e.g., Battery Energy Storage Systems or BESS), and transformer design characteristics—such as how effectively local infrastructure dampens fast fluctuations (“ripples”) in instantaneous demand. While these additional layers of system modelling are outside the scope of the present study, the proposed method provides a foundational building block for such analyses and could be integrated into broader optimisation frameworks. These aspects warrant dedicated treatment in future work.

We have rewritten the Discussions section to include the following:

The growing prevalence of EVs and the increasing demand for rapid, reliable public charging infrastructure necessitate advanced predictive tools to accurately model charging patterns and durations, in order to enhance user experience, to facilitate the efficient management and planning of charging systems, and to reduce the impact of EV charging on the grid. This work introduces a deep learning framework that predicts DC fast charging profiles at scale across multiple EV models, a range of charger power ratings, and diverse real world contexts. The model is trained on a dataset of 909,135 real-world DCFC charging sessions from the UK and Germany, achieving 95% accuracy with an absolute error of less than one minute when predicting charging durations using six data points (power as a function of SoC) collected within five minutes. These results demonstrate the model’s robustness and its ability to generalise across different EV models and charging scenarios.

The model is designed for practical deployment. It operates in an event-driven fashion, updating predictions as new SoC readings are received, typically every one minute. Inference is computationally efficient: a single forward pass over 1,024 sessions completes in approximately 100 milliseconds on standard GPU hardware (Table 3), corresponding to a throughput of over 10,000 sessions per second. This is several orders of magnitude faster than the rate at which new input data becomes available, confirming that the method is capable of real-time operation even at scale.

Crucially, the model relies solely on standard charging session data available at the charger, namely charging power, SoC, and charger metadata. For optimal performance, additional covariates such as estimated battery capacity and ambient temperature should be used when available, and can be inferred or measured locally during the charging session. However, the model remains capable of delivering robust predictions in their absence, maintaining usability in settings where such information is incomplete or unavailable. This design supports broad applicability across diverse EV and infrastructure configurations.

The proposed method enables a range of applications across both system-level and consumer-facing domains. For distribution and transmission system operators (DSOs and TSOs), accurate near-term forecasts of charging behaviour based on minimal early-session data can facilitate more precise demand prediction and flexible load management. This is particularly relevant for grid-constrained areas with high EV adoption, where real-time profiling at scale can support congestion mitigation, dynamic tariff design, and anticipatory grid services such as demand shifting or curtailment strategies. On the consumer side, the method provides the technical foundation for features such as real-time

charging duration estimates, adaptive pricing notifications, and enhanced queueing logic at high-traffic charging sites. These use cases benefit from the model's ability to operate on a session-by-session basis without relying on persistent vehicle identifiers or proprietary internal vehicle data.

While the charging profile prediction model has been shown to be able to generalise to session data collected in the Netherlands, which is a geographically distinct region from the training data also within north western Europe, full adaptation to radically different markets or climate zones may require additional training data. However, the model architecture and self-supervised learning workflow are sufficiently flexible to accommodate such extensions. Future work could include cross-session user modelling, personalised prediction, or behavioural inference. However, such directions would raise new privacy considerations, particularly if persistent identifiers or richer metadata are introduced. In these cases, privacy-preserving techniques such as differential privacy or federated learning may be necessary to ensure appropriate handling of both user-level data and commercially sensitive infrastructure information. Further gains may be realised through model-based optimisation, informing real-time decisions such as charger assignment, load scheduling, or dynamic pricing based on predicted charging behaviour. The charging profile prediction model may also serve as a forward simulator within reinforcement learning frameworks for smart charging control. More advanced uncertainty quantification techniques could further improve transparency in safety or cost sensitive applications.

Comment 9

You address the application to different geographic regions. Mention here again (and also this is missing and should be included in the Abstract) the region of the charging profiles for training. Elaborate more on what "different climates, behaviors, infrastructure" are?

Author response to comment 9: Thank you for your comment.

We have made clear in the discussion section that the region of the charging profiles for training is the UK and Germany. We have included an additional test dataset from the Netherlands and tested the performance of our model trained entirely on UK and Germany data on it. The results are shown in Figure 8 (Figure 5f in the revised manuscript). It should be noted that the Netherlands data lacks information on estimated battery capacity, so the model tested on it was a variant trained without that information. The results showed that the performance on the Netherlands data is comparable to that of the two test sets, showing to some extent the generalisability of our model across northwestern Europe.

Figure 8: Model evaluation on held-out test sessions from the Netherlands.

We mentioned "different climates, charging behaviours and infrastructure" because the proposed model is designed to function reliably within the operational domain captured by the training data, which is close to one million EV-charging sessions collected across DC chargers in the UK and Germany, which together reflect the dominant real-world usage patterns in those markets. As is typical for modern machine learning systems, the proposed predictive model is not intended to extrapolate to entirely novel contexts (e.g., completely new battery chemistries, fundamentally different charging standards, extreme polar or desert climate etc.). In a real-world deployment scenario, the emergence of such distributional shifts would naturally lead to performance degradation, until those new behaviours are incorporated into the training set through periodic retraining. At that point, the system is re-aligned with the evolving operational domain. The model is therefore not designed to extrapolate far beyond its training distribution, but rather to adapt over time as new data become available.

Comment 10

Future work should be addressed.

Author response to comment 10: Thank you for your comment. We have now addressed future work in the discussion section.

We have added the following to the Discussions section:

Future work could include cross-session user modelling, personalised prediction, or behavioural inference. However, such directions would raise new privacy considerations, particularly if persistent identifiers or richer metadata are introduced. In these cases, privacy-preserving techniques such as differential privacy or federated learning may be necessary to ensure appropriate handling of both user-level data and commercially sensitive infrastructure information. Further gains may be realised through model-based optimisation, informing real-time decisions such as charger assignment, load scheduling, or dynamic pricing based on predicted charging behaviour. The charging profile

prediction model may also serve as a forward simulator within reinforcement learning frameworks for smart charging control. More advanced uncertainty quantification techniques could further improve transparency in safety or cost sensitive applications.

References

- [1] Junzhe Shi, Min Tian, Sangwoo Han, Tung-Yan Wu, and Yifan Tang. Electric vehicle battery remaining charging time estimation considering charging accuracy and charging profile prediction. *Journal of Energy Storage*, 49:104132, 2022.
- [2] Shengyu Tao, Chongbo Sun, Shiyi Fu, Yu Wang, Ruifei Ma, Zhiyuan Han, Yaojie Sun, Yang Li, Guodan Wei, Xuan Zhang, Guangmin Zhou, and Hongbin Sun. Battery cross-operation-condition lifetime prediction via interpretable feature engineering assisted adaptive machine learning. *ACS Energy Letters*, 8(8):3269–3279, 2023.
- [3] Shengyu Tao, Haizhou Liu, Chongbo Sun, Haocheng Ji, Guanjun Ji, Zhiyuan Han, Runhua Gao, Jun Ma, Ruifei Ma, Yuou Chen, Shiyi Fu, Yu Wang, Yaojie Sun, Yu Rong, Xuan Zhang, Guangmin Zhou, and Hongbin Sun. Collaborative and privacy-preserving retired battery sorting for profitable direct recycling via federated machine learning. *Nature Communications*, 14(1):8032, 2023.
- [4] Shengyu Tao, Ruifei Ma, Yuou Chen, Zheng Liang, Haocheng Ji, Zhiyuan Han, Guodan Wei, Xuan Zhang, and Guangmin Zhou. Rapid and sustainable battery health diagnosis for recycling pretreatment using fast pulse test and random forest machine learning. *Journal of Power Sources*, 597:234156, 2024.
- [5] Shengyu Tao, Ruifei Ma, Zixi Zhao, Guangyuan Ma, Lin Su, Heng Chang, Yuou Chen, Haizhou Liu, Zheng Liang, Tingwei Cao, Haocheng Ji, Zhiyuan Han, Minyan Lu, Huixiong Yang, Zongguo Wen, Jianhua Yao, Rong Yu, Guodan Wei, Yang Li, Xuan Zhang, Tingyang Xu, and Guangmin Zhou. Generative learning assisted state-of-health estimation for sustainable battery recycling with random retirement conditions. *Nature Communications*, 15(1):10154, 2024.
- [6] Cheng Qian, Binghui Xu, Liang Chang, Bo Sun, Qiang Feng, Dezhen Yang, Yi Ren, and Zili Wang. Convolutional neural network based capacity estimation using random segments of the charging curves for lithium-ion batteries. *Energy*, 227:120333, 2021.
- [7] Jinpeng Tian, Rui Xiong, Weixiang Shen, Jiahuan Lu, and Xiao-Guang Yang. Deep neural network battery charging curve prediction using 30 points collected in 10 min. *Joule*, 5(6):1521–1534, 2021.
- [8] Laisuo Su, Shuyan Zhang, Alan J. H. McCaughey, B. Reeja-Jayan, and Arumugam Manthiram. Battery charge curve prediction via feature extraction and supervised machine learning. *Advanced Science*, 10(26):2301737, 2023.
- [9] Yupeng Lin, Qiuyang Liu, Yuanlong Chen, Chunyu Wang, Junjie Wang, and Lingling Zhao. Deep neural network battery charging curve prediction incorporating external information. *Journal of Power Sources*, 600:234189, 2024.
- [10] Qian Huo, Zhikai Ma, Xiaoshun Zhao, Tao Zhang, and Yulong Zhang. Bayesian network based state-of-health estimation for battery on electric vehicle application and its validation through real-world data. *IEEE Access*, 9:11328–11341, 2021.
- [11] Lingjun Song, Keyao Zhang, Tongyi Liang, Xuebing Han, and Yingjie Zhang. Intelligent state of health estimation for lithium-ion battery pack based on big data analysis. *Journal of Energy Storage*, 32:101836, 2020.
- [12] Qiushi Wang, Zhenpo Wang, Lei Zhang, Peng Liu, and Zhaosheng Zhang. A novel consistency evaluation method for series-connected battery systems based on real-world operation data. *IEEE Transactions on Transportation Electrification*, 7(2):437–451, 2021.
- [13] Jingzhao Zhang, Yanan Wang, Benben Jiang, Haowei He, Shaobo Huang, Chen Wang, Yang Zhang, Xuebing Han, Dongxu Guo, Guannan He, and Minggao Ouyang. Realistic fault detection of li-ion battery via dynamical deep learning. *Nature Communications*, 14, 09 2023.
- [14] Gabriele Pozzato, Anirudh Allam, Luca Pulvirenti, Gianina Alina Negoita, William A. Paxton, and Simona Onori. Analysis and key findings from real-world electric vehicle field data. *Joule*, 7(9):2035–2053, 2023.

Authors' Response to Reviewer 1

General Comments. The reviewer appreciated authors' efforts in addressing the comments and the quality of paper has been improved. However, the reviewer finds some issues to be carefully addressed before publication.

Author response to general comment: Thank you for your comments. We have responded to them item by item as follows.

Comment 1

While the reviewer acknowledges that the proposed model is designed to operate with a small segment of input data, the use of the term "minimal data" in the paper title feels overly strong and somewhat vague. It is recommended to soften this claim to better reflect the practical limitations and context of the approach.

Author response to comment 1: Thank you for your comment.

We appreciate the suggestion and have revised the title to "Artificial intelligence predicts real-world EV DC charging profiles and durations." This removes the phrase "with minimal data" while the manuscript text retains a clear explanation that the method operates with a single initial input point and updates predictions as more data arrive.

Comment 2

The reviewer found that Energy Environ. Sci., 2025,18, 7413-7426 (Immediate remaining capacity estimation of heterogeneous second-life lithium-ion batteries via deep generative transfer learning) has a similar motivation using minimal data to make predictions of battery states using a VAE architecture. Please consider discussing this in the introduction to enrich the depth of this paper.

Author response to comment 2: Thank you for your comment and for recommending this relevant article from the recent literature; we have indeed decided to cite it in the revised manuscript.

We have edited the introduction to reflect these changes:

Recent advances in data-driven modelling of lithium-ion battery systems have increasingly addressed challenges associated with random, scarce, and heterogeneous data using sophisticated machine learning techniques [1, 2, 3, 4, 5]. These studies, often based on real-world datasets, have achieved strong performance in tasks such as state of health estimation, lifetime prediction, and retired battery sorting, and represent meaningful progress toward the development of robust and generalisable battery analytics.

Comment 3

The reported accuracy of 90% is presented without sufficient contextual justification. Without a clear explanation of the target application’s tolerance for error, this number may lack practical significance. It is advised to either clarify why this level of accuracy is acceptable or avoid highlighting it as a standalone metric.

Author response to comment 3: Thank you for your comment, we have revised the manuscript to clarify the context for the 90% single-point accuracy. This figure represents the lower bound of performance when only one initial data point is available; predictions continue to update as more points are incorporated, reaching higher accuracy levels (Figure 4 in the revised manuscript). We now make explicit that performance is evaluated using multiple metrics across input lengths from 1 to 15 Power-SoC points, with the 90% case being one element within this broader evaluation scheme. In DC fast-charging operations, even this early accuracy can support time-sensitive decisions such as anticipating charger turnover or estimating completion time, with subsequent updates further refining predictions.

We have made the following changes to the results section, Time to reliable prediction operationally post-plug-in subsection:

The model’s proficiency in both predicting the charging profiles and estimating the corresponding charging times is evaluated for the two test sets and reported in Figure 4a, where the median predicted charging profiles and their associated charging times are compared against ground truth values. It can be seen that while the model performed marginally better in test set one than in test set two, the difference is relatively small. Overall, the model achieved an average accuracy of approximately 90% for both test sets, with an average error of less than 2.5 minutes, when used to predict charging times based on only a single data point from the charging profile. This value is the lower bound of the model’s performance in predicting charging time using relative error, with performance also assessed using absolute-error metrics such as MAE in charging time and charging curve accuracy across input lengths from 1 to 15 points. Accuracy increases progressively as more data become available, and even this lower-bound accuracy can enable operationally useful actions such as anticipating charger availability, assessing session completion within certain time windows, and providing early completion time estimates to drivers.

Comment 4

To ensure that the reported performance is not the result of overfitting, the authors should consider including a random shuffling experiment or adding training vs. validation loss plots. This would help assess whether the model’s high accuracy is generalizable or confined to the specific dataset used.

Author response to comment 4: Thank you for your comment. We have already done both, and have now made this clear in the revised manuscript. Specifically, we conducted three independent training runs with different random seeds, different random initialisation of model weights and optimiser states, different data shuffling, and different train-validation splits. In Figure 1 we show the training loss (quantile loss with random

masking) as well as validation losses (quantile loss at different levels of masking).

We have edited the supplementary information to reflect these changes:

Figure 1 shows the training and validation loss curves for the TFT model from three separate training runs, each initialised with a different random seed. The differing seeds resulted in variations in weight initialisation, data shuffling, and train–validation splits. All models were trained to convergence using a reduce-on-plateau learning rate scheduler, which lowered the learning rate when improvements in validation loss plateaued. Model selection was based on the sum of validation losses across the different input point levels.

Figure 1: Training and validation loss curves of the charging profile prediction model with the TFT architecture. The model was trained using the quantile loss with three random seeds with different random initialisation and data shuffling. **a** Training losses where a random portion of the charging curves are masked for training. **b** Validation losses with model performance evaluated using fixed input point counts of 1, 5, 10, and 15.

Comment 5

Although the paper states that the test data was completely withheld during training, it is possible that the testing samples lie within the subspace of the training distribution. Since Wasserstein distance alone may not offer intuitive insight into data separation, it would be helpful to include a visualization of the data distributions (e.g., PCA or t-SNE plots) to support the claim that the test set truly represents unseen conditions.

Author response to comment 5: Thank you for your comment.

Because the dataset is large ($> 7 \times 10^5$ training sessions and $\sim 4.5 \times 10^4$ per test set), raw t-SNE scatter is saturated by overplotting. We therefore use density ratio maps of the t-SNE embeddings, computing $\log(\text{density}_{\text{test}}/\text{density}_{\text{train}})$ on a common grid under a shared colour scale.

We have included the following in the supplementary information section:

Figure 2: t-SNE latent space density ratio maps comparing each test set with the training set under a shared colour scale. Colour intensity reflects the magnitude of the relative density difference with respect to the training set; hue encodes direction (blue indicates under-representation in the test split, red indicates over-representation). **Left:** test set one vs training shows mostly small, diffuse deviations near zero. **Right:** test set two vs training shows coherent regions of deficit and pockets of excess, consistent with a temporal distribution shift.

To complement the permutation analysis, distributional differences in the β -VAE latent space were visualised with t-SNE density ratio maps and shown in Figure 2. Latent vectors from the training set and from each test set were embedded to two dimensions with t-SNE. Densities were estimated on a common grid and lightly smoothed. For each bin the log ratio

$$\log\left(\frac{\text{density}_{\text{test}}}{\text{density}_{\text{train}}}\right)$$

was computed, and bins with insufficient support were masked. A single symmetric colour scale was applied across both panels.

Authors' Response to Reviewer 2

General Comments. I co-reviewed this manuscript with one of the reviewers who provided the listed reports. This is part of the Nature Communications initiative to facilitate training in peer review and to provide appropriate recognition for Early Career Researchers who co-review manuscripts.

Author response to general comment: Thank you for your time and involvement in the review process. We appreciate your contribution through the co-review and hope the revised manuscript meets expectations.

Comment 1

The data and code are not fully disclosed due to claimed restrictions.

Author response to comment: Thank you for your comment.

As in the previous rounds, we continue to provide the full anonymised dataset, pre-trained models, and accompanying codebase under confidentiality. The source code is submitted via the journal portal, and the dataset and models remain available at the same Google Drive links as before. We wish to clarify for the record that these materials comprise the complete dataset and code used in this study, made available in full to the reviewers for their assessment.

- Dataset: <https://drive.google.com/file/d/1SDGQvv2yn2tzmMbBqAchM3tBV9pQXuLn/view?usp=sharing>
- Pre-trained models: https://drive.google.com/file/d/13RtcQ0nj6r11YUAZ6bsU_5nwnYWX-C1E/view?usp=sharing

Authors' Response to Reviewer 4

General Comments. Dear authors,

I believe that most the the criticism I had has been addressed and the readability as well as scientific quality of the manuscript has been significantly improved.

I have major issue that I have already addressed during the first round of the review - maybe not directly: The visualizations include a lot of information which I still don't find necessary and overall disruptive to the communication of your research. For example: d-f in Figure 2 - These are not discussed within the text. Just the visualization is included. Removing these subfigures could give way better opportunity to explain a-c. The arrangement of the figures is rather confusing and just looking at the visualizations there is no natural visual guidance. I understand the necessity to display the information but I believe that for this interdisciplinary journal, intuitive visualizations are essential.

Author response to general comment: Thank you for your comment. We have removed Figure 2 subplots d-f as you requested.

References

- [1] Shengyu Tao, Chongbo Sun, Shiyi Fu, Yu Wang, Ruifei Ma, Zhiyuan Han, Yaojie Sun, Yang Li, Guodan Wei, Xuan Zhang, Guangmin Zhou, and Hongbin Sun. Battery cross-operation-condition lifetime prediction via interpretable feature engineering assisted adaptive machine learning. *ACS Energy Letters*, 8(8):3269–3279, 2023.
- [2] Shengyu Tao, Ruifei Ma, Zixi Zhao, Guangyuan Ma, Lin Su, Heng Chang, Yuou Chen, Haizhou Liu, Zheng Liang, Tingwei Cao, Haocheng Ji, Zhiyuan Han, Minyan Lu, Huixiong Yang, Zongguo Wen, Jianhua Yao, Rong Yu, Guodan Wei, Yang Li, Xuan Zhang, Tingyang Xu, and Guangmin Zhou. Generative learning assisted state-of-health estimation for sustainable battery recycling with random retirement conditions. *Nature Communications*, 15(1):10154, 2024.
- [3] Shengyu Tao, Ruifei Ma, Yuou Chen, Zheng Liang, Haocheng Ji, Zhiyuan Han, Guodan Wei, Xuan Zhang, and Guangmin Zhou. Rapid and sustainable battery health diagnosis for recycling pretreatment using fast pulse test and random forest machine learning. *Journal of Power Sources*, 597:234156, 2024.
- [4] Shengyu Tao, Haizhou Liu, Chongbo Sun, Haocheng Ji, Guanjun Ji, Zhiyuan Han, Runhua Gao, Jun Ma, Ruifei Ma, Yuou Chen, Shiyi Fu, Yu Wang, Yaojie Sun, Yu Rong, Xuan Zhang, Guangmin Zhou, and Hongbin Sun. Collaborative and privacy-preserving retired battery sorting for profitable direct recycling via federated machine learning. *Nature Communications*, 14(1):8032, 2023.
- [5] Shengyu Tao, Ruohan Guo, Jaewoong Lee, Scott Moura, Lluc Canals Casals, Shida Jiang, Junzhe Shi, Stephen Harris, Tongda Zhang, Chi Yung Chung, Guangmin Zhou, Jinpeng Tian, and Xuan Zhang. Immediate remaining capacity estimation of heterogeneous second-life lithium-ion batteries via deep generative transfer learning. *Energy Environ. Sci.*, 18:7413–7426, 2025.